# ELLA: EMBODIED LIFELONG LEARNING AGENTS WITH NON-PARAMETRIC MEMORY

## ABSTRACT

Situated within human society, embodied agents are continuously exposed to diverse streams of information, ranging from visual observations to natural language interactions. A central challenge is enabling them to learn from and effectively leverage this information over extended periods. To address this, we introduce *Ella*, an embodied lifelong learning agent designed to accumulate experiences and acquire knowledge across hours of social interaction in a 3D open world. At the core of *Ella*'s capabilities is a structured, non-parametric, long-term multi-modal memory system that stores, updates, and retrieves information effectively. It consists of a name-centric semantic memory for organizing acquired knowledge and a spatiotemporal episodic memory for capturing multimodal experiences. By integrating foundation models with this non-parametric memory system, *Ella* retrieves relevant information for decision-making, plans daily activities, builds social relationships, and evolves autonomously while coexisting with other intelligent beings in the open world. We conduct capability-oriented evaluations in a dynamic 3D open world where 15 agents engage in social activities for days and are assessed with a suite of unseen controlled evaluations. Experimental results show that *Ella* can influence, lead, and cooperate with other agents well to achieve goals, showcasing its ability to learn effectively through observation and social interaction. Our findings highlight the transformative potential of combining non-parametric memory systems with foundation models for advancing embodied intelligence. More videos can be found at https://ellaiclr2026.github.io/Ella.

## 1 INTRODUCTION

It's a long-standing goal to create intelligent beings capable of survival in the human community (Gan et al., 2021; Li et al., 2023a; Puig et al., 2024), which requires lifelong learning in an open and social world. The embodied agents must accumulate experiences, including visual observations and social interactions with other intelligent beings, such as conversations; and acquire knowledge from these multi-modal experiences, build new concepts of objects, agents, and events, and identify the connections among these concepts.

With the rapid advancement of Foundation Models (OpenAI, 2023; Ravi et al., 2024; Guo et al., 2025), a surge of powerful agents has emerged (Sumers et al., 2023). These range from agents operating solely in the text domain (Gur et al., 2023; Shinn et al., 2024) to multi-modal agents capable of controlling screens (Hong et al., 2024), playing games (Wang et al., 2023a;b), and even functioning as robots in the physical world (Ahn et al., 2022; Huang et al., 2023b; Du et al., 2023). Despite these advancements, one crucial component remains underexplored in current agent research: long-term memory. Humans organize accumulated experiences in Episodic Memory (Tulving, 1972; 1983; Nuxoll & Laird, 2007) and acquired knowledge in Semantic Memory(Lindes & Laird, 2016), enabling them to make long-term plans and exhibit higher-level cognitive capabilities (Laird, 2022; Tenenbaum et al., 2011). In contrast, current work in embodied agents is limited to constrained spatial regions (primarily indoor spaces) and brief temporal scales (seconds for robotic manipulation or minutes for navigation tasks). For agents to thrive in an ever-evolving world, it is essential to develop a long-term memory system that supports learning new concepts and forming new relationships. However, directly fine-tuning the parameters of large foundation model-based agents has been shown to suffer from catastrophic forgetting (Huang et al., 2024). Developing non-parametric memory and effective retrieval algorithms has been a practical alternative for continual learning with LLMs (Gutiérrez et al.,

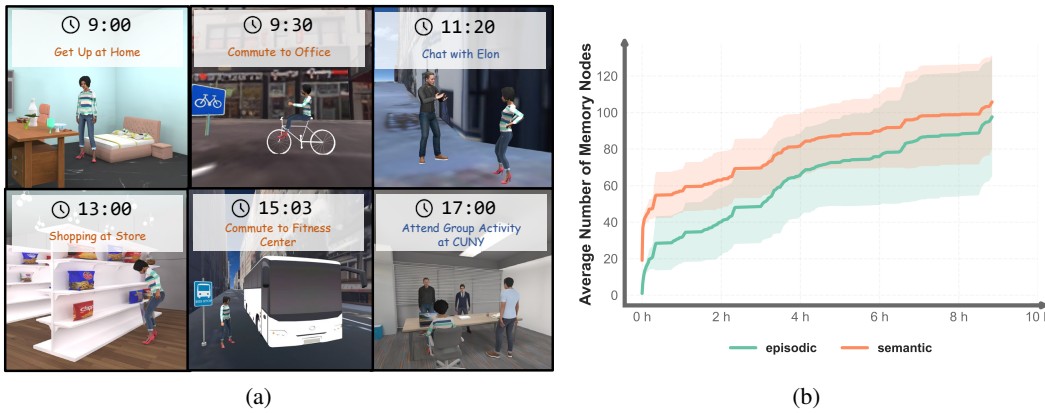

(a)                 (b)

Figure 1: (a) Embodied agents require lifelong learning to accumulate experiences and acquire knowledge through everyday visual observation and social interaction within a community in a 3D open world. (b) *Ella* self-evolves by growing episodic and semantic memory over time.

2025). For Agents, Generative Agents (Park et al., 2023) introduced a textual temporal episodic memory, assuming oracle perception in a sandbox 2D environment. Similarly, Voyager (Wang et al., 2023a) designed a single agent with long-term procedural memory, enabling it to acquire new skills in Minecraft through oracle perception and self-training. However, the challenge of constructing effective lifelong memory systems for embodied agents in an open and social world—where they must learn from visual observations and engage in social interactions with other intelligent beings, as illustrated in Figure 1a—remains largely unexplored.

In this work, we propose to build a non-parametric long-term memory system that can store, update, and retrieve information effectively. Borrowing the concepts from psychology and cognitive neuroscience (Tulving, 1972), we construct the long-term memory in two forms: a name-centric semantic memory with a hierarchical scene graph and knowledge graph to organize acquired knowledge, and a spatiotemporal episodic memory to capture the agent's multi-modal experiences. We present *Ella*, an embodied lifelong learning agent that can accumulate experiences and acquire knowledge effectively through visual perception and social interaction with other agents within a community in an open 3D world, by integrating this non-parametric memory with foundation models. To plan robustly and behave consistently through days of social life, *Ella* adopts a planning-reaction framework where it first retrieves related memory to make a structured daily schedule, then updates the memory with new visual observations and social interactions, and make reactions to the new context, which could be to revise the schedule, interact with the environment, or engage in social interactions.

We simulate *Ella* and other baseline agents in *Virtual Community* (Zhou et al., 2025), an open world simulation platform for multi-agent embodied AI, featuring large-scale community scenes with realistic physics and renderings. Unlike traditional task-oriented evaluations for agents, assessing high-level cognitive capabilities in a lifelong setting is more critical (Crosby et al., 2019). To this end, we first simulate 15 agents for 9 hours (with an observation and control frequency of 1 second), representing their first day in the community. During this phase, agents must plan their day based on their unique characteristics and acclimate to the environment and other agents. Then we test the agents with unseen controlled evaluations: *Influence Battle* and *Leadership Quest*, where the agents work in groups to persuade others to attend their party at a specific location despite conflicting schedules or lead their group to prepare for an activity under resource constraints. Experimental results across three communities show that *Ella* demonstrates advanced cognitive abilities including social reasoning and leadership, compared to baselines with only short-term memory or episodic memory, showcasing its ability to learn effectively through visual observation and social interaction. In sum, our contribution includes:

- We build a non-parametric long-term memory with name-centric semantic memory and spatiotemporal episodic memory for embodied lifelong learning in an open and social world.

- We introduce ***Ella***, an embodied lifelong learning agent that can self-evolve through visual observation and social interaction by integrating long-term memory with foundation models.

- We conduct capability-oriented experiments in a 3D open world with 15 agents where ***Ella*** demonstrates superior cognitive abilities with a more advanced long-term memory.

Figure 2: **An example community of 15 agents and 4 social groups in New York.** The character and observation of agent *Elizabeth Mensah* are shown on the right.

## 2 RELATED WORK

### 2.1 EMBODIED SOCIAL INTELLIGENCE

Social intelligence has been widely studied in embodied multi-agent environments (Zhou et al., 2025; Lowe et al., 2017; Carroll et al., 2019; Amato et al., 2019; Bard et al., 2020; Jain et al., 2020; Puig et al., 2021; Tsoi et al., 2020; Puig et al., 2023; Wen et al., 2022; Szot et al., 2023; Zhang et al., 2023; Li et al., 2019), while one branch focuses on simplified symbolic or game-like environments (Samvelyan et al., 2019; Suarez et al., 2019; Jaderberg et al., 2019; Baker et al., 2020; Niu et al., 2021; Sharon et al., 2015; Yu et al., 2024), often ignoring the challenges present in an open world, including perception and diverse personalities of agents. Specifically, generative agents (Park et al., 2023) developed a unified temporal memory, demonstrating the robust simulation of human-like agents within a symbolic community. Following this line of research, a series of works have explored socially intelligent agents within text-based sandbox environments (Li et al., 2023b; Zhou et al., 2024; Liu et al., 2024a; Chen et al., 2024; Liu et al., 2024b;c; Dai et al., 2024). The other branch, including works on human-robot interaction (Gombolay et al., 2015; Goodrich et al., 2008; Bobu et al., 2023; Dautenhahn, 2007; Nikolaidis et al., 2015; Rozo et al., 2016; Losey et al., 2022; Natarajan & Gombolay, 2020; Lasota et al., 2017), focuses on real-world domains but is limited to specific task settings. Different from above, we explore embodied social intelligence within a community in an open 3D world, featuring expansive spatial regions and a temporal scale spanning multiple days.

### 2.2 AGENT MEMORY

Memory has been studied for a long time in AI, especially related to cognitive architectures (Weston et al., 2014; Lindes & Laird, 2016; Sumers et al., 2023). However, most modern agent architecture primarily assumes a temporal memory due to the constraints of specific domains or the limited time horizon for which the agent is designed. A visual memory as a type of semantic memory has been implemented using various structures in computer vision, including voxels (Chaplot et al., 2020; Blukis et al., 2022; Min et al., 2022; Ramakrishnan et al., 2022), scene graphs (Li et al., 2022; Rana et al., 2023; Kurenkov et al., 2023; Gu et al., 2024b), Octrees (Hornung et al., 2013; Zhang et al., 2018; Asgharivaskasi & Atanasov, 2023; Zheng et al., 2023), or implicit continuous representations (Shafiullah et al., 2022; Gadre et al., 2022; Huang et al., 2023a; Gadre et al., 2023). Recently, several works have explored agent memory for longer time horizons. (Kurenkov et al., 2023; Yang et al., 2024) and (Zhou et al., 2023) introduce updating mechanisms for scene graph-based memory, adapting it to long-term tasks. (Wang et al., 2023a) and (Li et al., 2024) develop procedural memory tailored for specific game environments to support long-term planning. (Jiang et al., 2024) proposes long-term memory with a graph-based structure to enable self-evolution in LLM tasks. (Wang et al., 2024) further integrates long-term and short-term memory to address long-horizon tasks within household environments. Another line of work studies how to better retrieve knowledge from external data sources to help Large Language Models answer questions (Borgeaud et al., 2022; Gutiérrez et al., 2024; 2025; Packer et al., 2023; Han et al., 2024; Shi et al., 2024; Yasunaga et al., 2023). However, none of the above have studied how to build a long-term memory system that could learn from both visual observations of the environment and social interactions with other agents, which we tackled with a dual-form structured memory and foundation models.

## 3 PROBLEM SETTING

In our setting, $n$ agents with unique visual appearance $v_i$ and character profile $c_i$ inhabit an open, socially interactive world $W$, forming $k$ social groups, as illustrated in Figure 2. Each character's profile is defined by basic attributes such as name, age, occupation, values (Schwartz, 2012), hobbies, lifestyle, and current goals within the community. These attributes guide the agent's daily decision-making. Social groups consist of a subset of agents selected based on character compatibility, and are defined by a group name, a detailed textual description, and a designated physical location for group activities. These groups connect the agents into a cohesive community, allowing rich and complex social interactions grounded in the 3D environment. Each agent is initialized with partial knowledge about the world, including known places and familiar agents, such as their residence and fellow group members, based on their characters' profile. The simulation runs at a fine temporal resolution of one second per step, during which each agent receives an observation $o_i$ including posed RGB and depth images, as well as dialogue content from nearby agents. Communication is spatial-constrained: agents can only engage in conversation if they are within a threshold distance $\theta_s$, mimicking realistic spatial constraints on verbal interactions. Every second, agents execute an action $a_i$ to interact with the environment or other agents. During controlled evaluations, intervention occurs solely through modifications to agents' community goals. Agents are required to make optimal decisions $a_i$ based on their updated character profiles $c_i$ and incoming observations $o_i$.

## 4 ELLA: EMBODIED LIFELONG LEARNING AGENT

To enable the embodied agents to continually learn within a community in a 3D open world, robust and efficient long-term memory is the key. Many study finds that endorsing ever-evolving long-term memory by tuning the parameters of the LLMs faces the difficulty of catastrophic forgetting (Cohen et al., 2024; Gu et al., 2024a), while the non-parametric approach of building a knowledge base and retrieving new information from it avoids such challenges (Gutiérrez et al., 2025). Borrowing the concepts from psychology and cognitive neuroscience (Tulving, 1972), we build non-parametric long-term memory in two forms: name-centric semantic memory (Section 4.1) and spatiotemporal episodic memory (Section 4.2). Then in Section 4.3, we introduce how we leverage the foundation models to integrate this memory system to facilitate the agent's everyday planning and social interactions.

### 4.1 NAME-CENTRIC SEMANTIC MEMORY

Semantic memory stores facts about the agent and world, which is continually updated while the agent interacts with the world and other agents. Different from language agents, which normally take external databases like Wikipedia as a form of knowledge to help reasoning (Sumers et al., 2023; Lewis et al., 2020; Borgeaud et al., 2022), embodied agents need knowledge grounded in the environment they inhabit. We organize the different types of knowledge in a name-centric way and connect the related ones into a graph as shown in Figure 3 (a). Specifically, we build a hierarchical scene graph on the fly to serve as a spatial memory to help the agent navigate the visual world. The semantic memory is updated whenever there is a new visual observation made or a conversation finished, as introduced in Section 4.3.3.

### 4.1.1 HIERARCHICAL SCENE GRAPH AS SPATIAL MEMORY

Maintaining a spatial memory of the surrounding world is vital for embodied agents to act in a 3D world. To serve this purpose, we incrementally build a hierarchical scene graph (Hughes et al., 2022; Gu et al., 2024b) on the fly as shown in Figure 3 (a).

**Volume Grid Layer** Given posed RGB and depth observation, we first project them to 3D space and represent them in volume grid representations to act as low-level geometric memory. We then obtain an occupancy map based on it to facilitate navigation while avoiding obstacles in the 3D world. We divided the entire map into blocks of 0.5m × 0.5m and subdivided each block into smaller cells of 0.1m × 0.1m. We identified the lowest position within each small cell that could accommodate a person. A cell was classified as containing an obstacle if the height difference between this position and any of its neighboring cells exceeded 0.5m.

**Object Layer** Taking inspiration from previous works (Gu et al., 2024b; Maggio et al., 2024), we employ a multi-stage perception pipeline to process RGB observations in an open world. Specifically, we utilize a combination of open-set vision models—including tagging, object detection, and segmentation—to form the perception module. This module extracts a sequence of semantically labeled masks $\langle m_i, \text{tag}_i \rangle$ as object candidates. Using depth and pose observations, each mask $m_i$ is projected into a 3D point cloud $p_i$, enabling the computation of geometric similarity $\text{sim}(p_i, p_j)$ between objects based on their spatial overlap. Additionally, we extract visual features $v_i$ for each

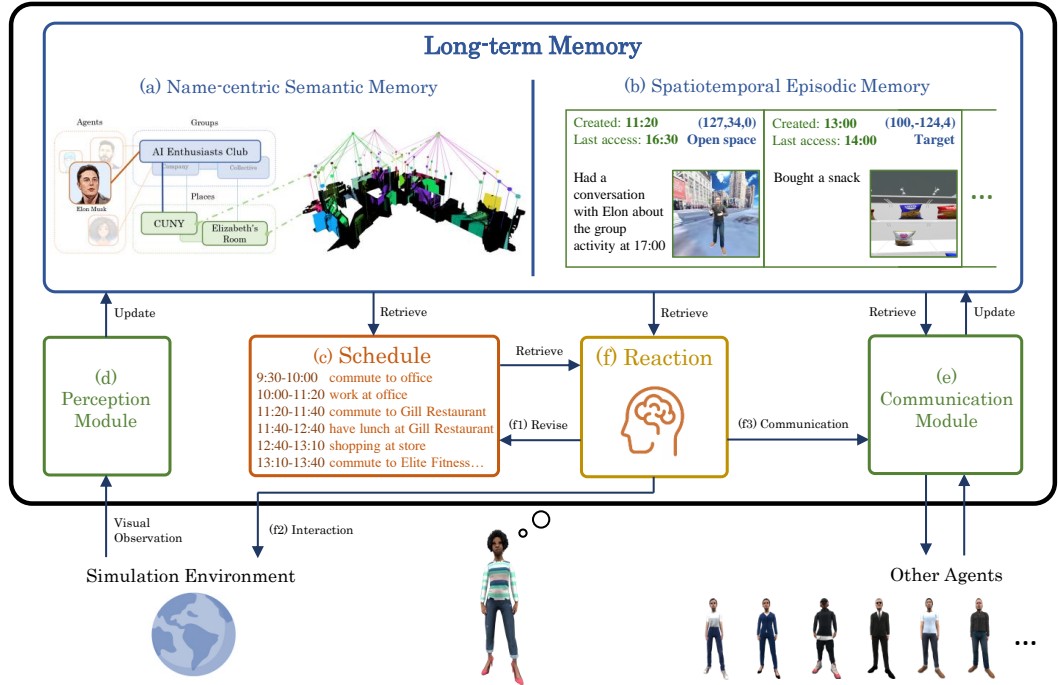

Figure 3: **Method Overview.** We build non-parametric long-term memory in two forms: (a) name-centric semantic memory organizes the knowledge in a name-centric graph including a hierarchical scene graph serving as the spatial memory; (b) spatiotemporal episodic memory stores the experience as a series of events consisting of time, location, and multimodal contents. (c) *Ella* first generates a daily schedule according to the knowledge and experiences retrieved from the long-term memory, (d) then updates the memory based on visual observations of the environment, and (e) social interactions with other agents and (f) makes reactions accordingly including (f1) revising the schedule, (f2) interacting with the environment, (f3) and engaging in a conversation.

object by encoding the corresponding cropped image. The detected object candidates from the current frame are then merged with existing objects based on similarity measurements. Unlike Gu et al. (2024b), we handle the additional complexity of dynamic objects such as agents and vehicles. Due to the relatively low perception rate (1 FPS), conventional tracking techniques are impractical. Instead, we rely on visual similarity to associate and merge dynamic objects across frames.

**Region Layer** We also implemented a region layer to further classify the buildings. First, we used the occupancy map and a breadth-first search to compute the Generalized Voronoi Diagram (GVD)(Hughes et al., 2022) of the map. For each point $p$ in the GVD, we determined the set $S = \arg\min\{\text{dist}(p, b) | b \in B\}$, where $B$ represents the set of all buildings. We then connected all buildings in $S$ with edges weighted by $\frac{1}{\text{dist}(p,s)^2}$, where $s \in S$. Finally, we connected all previously unconnected buildings by adding edges with zero weight, resulting in a complete graph. To group nodes connected by higher-weight edges, we applied spectral clustering, partitioning the graph into $\sqrt{|B|}$ regions. This clustering facilitated a more structured geometric partitioning of the buildings.

### 4.2 SPATIOTEMPORAL EPISODIC MEMORY

Episodic memory is responsible for storing personal experiences (Tulving, 1972; 1983). As noted by Mastrogiuseppe et al. (2019), episodic memory encodes not only when and what events occurred but also where they took place—highlighting the crucial role of spatial information. Unlike Park et al. (2023), our episodic memory module incorporates both temporal and spatial information, in addition to multi-modal content, enabling the agent to retrieve experiences relevant to its current location. Experiences are stored as a series of events, each composed of temporal attributes (event creation time and last access time), spatial attributes (event location and place), and content attributes (a textual description and a corresponding egocentric image), as illustrated in Figure 3(b).

**Retrieval** The episodic memory supports spatiotemporal retrieval. Given a query—comprising time, location, and content—all stored experience items are ranked based on the following three criteria:

*Spatial Proximity* measures the distance between the event location $p_e$ and the query location $p_q$.
$$\text{proximity}(e, q) = \frac{1}{\|\mathbf{p}_e - \mathbf{p}_q\| + \epsilon}$$

*Content Relevance* measures how well an event's content aligns with the given query by evaluating both textual and visual similarity. Specifically, we compute the cosine similarity between the encoded representations of the event and query, considering both their text descriptions $T$ and images $I$. The final relevance score is obtained by averaging these two similarities. $\text{Relevance}(e, q) = (cos(T_e, T_q) + cos(I_e, I_q))/2$

*Temporal Recency* is higher for events recently accessed. Following (Park et al., 2023), we model recency using an exponential decay function based on the time elapsed since the memory was last accessed. $\text{Recency}(e) = \exp{(t_e - t_q)}$

All three scores are then normalized to the range of $[0, 1]$ with min-max scaling and averaged as the final score, and the top $k$ events are retrieved.

### 4.3 PLANNING, REACTION, AND COMMUNICATION

With this structured long-term memory, *Ella* leverages foundation models to make efficient and robust everyday planning. Following Park et al. (2023), we adopt a planning and reaction framework with several modifications to facilitate efficient daily planning. *Ella* first generates an environment- and characters-grounded daily schedule according to the knowledge and experiences retrieved from the long-term memory, then updates the memory based on observations and makes reactions accordingly, including revising the schedule, engaging in a conversation, and interacting with the environment. A specific communication module is incorporated to generate the utterance to chat about, summarize the conversations, and extract knowledge from it. More details on the submodules are provided in Appendix B. All prompt templates are provided in Appendix C.

#### 4.3.1 DAILY SCHEDULE

At the start of each day, *Ella* will retrieve experience and knowledge from the long-term memory with a query of *"Things to consider for my schedule today."*, then use foundation models to generate the daily schedule. Different from Park et al. (2023), we generate the daily schedule in a structured manner directly with each activity represented with a start time, an ending time, an activity description, and the corresponding activity place. Specifically, we consider the required commute time between adjacent activities happening in different places explicitly, due to the actual cost of navigating in an expansive 3D environment. For example, commuting from the office to a party place may take more than 15 minutes on foot, without considering that the agent may miss the party if they planned to attend the party at the party's starting time. Figure 3 (c) shows a generated daily schedule for agent *Elisabeth Mansah*. The daily schedule may be revised later by the reaction module, given new experience and knowledge obtained from observations and social interactions during the day. There are four types of activities in the schedule: *commute, main, meal, and sleep*. For *commute* activity, *Ella* will make a commute plan based on available knowledge of the transit system and places in the community, then execute it by invoking the navigation submodule detailed in Appendix B. For activities of other types, a behavior planning module will be invoked to generate low-level action schedules based on the activity description and the retrieved information about the objects in the activity place. Please see Appendix B.1- B.3 for more details.

#### 4.3.2 REACTION

Upon receiving new observations, the agent first processes visual information and updates its semantic memory using the perception module introduced in Section 4.1.1. If new objects are detected or messages are heard, the agent invokes the reaction module. This module begins by retrieving relevant memories using the query *"Important things to react to."*, then use foundation models to reason about the character, current time, place, schedule, and retrieved memory and make one of the four choices: *revising the schedule, interacting with the environment, engaging in a conversation, or no reactions needed*, as illustrated in Figure 3 (f). Additionally, the reaction module is automatically triggered if the time elapsed since the last reaction exceeds $\theta_{react}$ seconds.

#### 4.3.3 COMMUNICATION

When the agent generates a reaction of *engaging in a conversation*, the communication module is revoked to generate the utterance by first retrieving the related knowledge and experience from the long-term memory with a query of the latest sentence in the conversation or *"Things to chat about with `conversation targets`"* if the agent is initiating a new conversation, then use foundation models to synthesize the appropriate utterance. When the conversation finishes, the communication module will summarize it and store the summarized conversation in episodic memory. *Ella* will also

Table 1: **Main results.** We report the show-up rate and the total number of conversations for **Influence Battle**, and the completion rate and the total number of conversations for **Leadership Quest**. + Oracle Perception assumes ground truth 2D segmentation. The best results are in **bold**. *Ella* achieves a higher show-up rate and completion rate across all three communities.

| | *Influence Battle* | | | | *Leadership Quest* | | | |
|---|---|---|---|---|---|---|---|---|
| | New York | London | Detroit | **Average** | New York | London | Detroit | **Average** |
| *CoELA* (Zhang et al., 2023) | 46.7, 57 | 20.0, 27 | 6.7, 17 | 24.5, 33.7 | 0.0, 72 | 0.0, 957 | 11.5, 625 | 3.8, 551.3 |
| *Generative Agents* (Park et al., 2023) | 40.0, 3 | 40.0, 0 | 20.0, 0 | 33.3, 1.0 | 8.3, 169 | 0.0, 55 | 16.7, 14 | 8.3, 79.3 |
| *+ Oracle Perception* | 46.7, 5 | 53.3, 153 | 26.7, 0 | 42.2, 52.7 | 4.2, 649 | 0.0, 5 | 16.7, 2 | 7.0, 218.7 |
| *Ella* (**Ours**) | 46.7, 12 | 66.7, 19 | 46.7, 15 | **53.4, 15.3** | 33.3, 15 | 26.7, 17 | 37.5, 14 | **32.5, 15.3** |
| *+ Oracle Perception* | 60.0, 11 | 60.0, 28 | 53.3, 17 | **57.8, 18.7** | 39.6, 87 | 35.0, 35 | 25.0, 26 | **33.2, 49.3** |

try to extract new knowledge it learned from the conversation by prompting a foundation model with some demonstration knowledge items, and use it to update the semantic memory.

## 5 EXPERIMENTS

### 5.1 EXPERIMENTAL SETUP

We instantiate our embodied social agents community in *Virtual Community*, an open world simulation platform for multi-agent embodied AI. We conducted experiments with 15 agents of unique characters in 3 different scenes and communities. The observation space consists of posed $512 \times 512$ RGB and depth images, messages received within range, and current states. To evaluate the effectiveness of the proposed non-parametric memory and high-level cognitive capabilities of the agents, we design our experiments in two stages as shown in Figure 8. In the first stage, 15 agents are simulated for 9 hours (34200 steps) for their first day in the community, during which the agents could familiarize themselves with the 600m * 600m scene and other agents and build memories. Then in the second stage, we test them with two controlled evaluations in the days following: ***Influence Battle*** and ***Leadership Quest***. In ***Influence Battle***, two of the four groups will be asked to organize a party at a specific place in 6 hours, and the members need to go around the city, find and invite agents outside of their group to attend the party. This evaluation tests the agents' capability to impact other agents by persuading them to attend the parties, which requires the capability of social reasoning, persuasion, and decision-making. In ***Leadership Quest***, each of the four groups is assigned a task to purchase several items from various stores in the city and return within 3 hours. One member from each group is designated as the leader and is the only one given full details of the task, while the remaining members are simply instructed to assist the leader. This controlled evaluation setting challenges the agent's leadership abilities, particularly in assigning sub-tasks based on the diverse personalities and resources of group members. More details on the tasks can be found in Appendix A.

**Metrics** We evaluate agents' capability to influence others with *show up rate*, the total number of agents showing up at any party organizing place during the 30-minute party time divided by the total number of agents; and *the total number of conversations* the organizing parties engaged in, reflecting the efficiency of the invitations. In ***Leadership Quest***, we measure the success of agents' leadership and cooperation by *average completion rate*, the number of fulfilled target items divided by the number of all target items averaged across all groups; and *the total number of conversations* reflecting the efficacy of the communications among the agents.

**Baselines** To the best of our knowledge, there hasn't been any embodied social agent framework supporting social interaction within a community with open-world 3D scenes. The most related methods are CoELA (Zhang et al., 2023), which only considered two agents within a constrained indoor scene for a specific task, and Generative Agents (Park et al., 2023), which assume oracle perception and use a predefined communication mechanism. We re-implemented these two methods in our setting as the baselines.

- *CoELA* (Zhang et al., 2023) is a cooperative embodied agent. We replace their perception module with ours since there isn't a pretrained 2D segmentation model available under our open-world setting. We provide the character description to replace the CoELA's task-specific description.
- *Generative Agent* (Park et al., 2023) is a believable simulacrum of human behavior with an uni-modality long-term memory. We adopt the perception module of Ella to convert visual observations into text descriptions and use the same occupancy map and a* algorithm for visual navigation.

**Implementation Details** For the perception module, we use open-set tagging model RAM++ (Huang et al., 2023c), object detection model GroundingDINO (Liu et al., 2023), and segmentation model SAM2 (Ravi et al., 2024). For the embedding models, we use CLIP (Radford et al., 2021) model

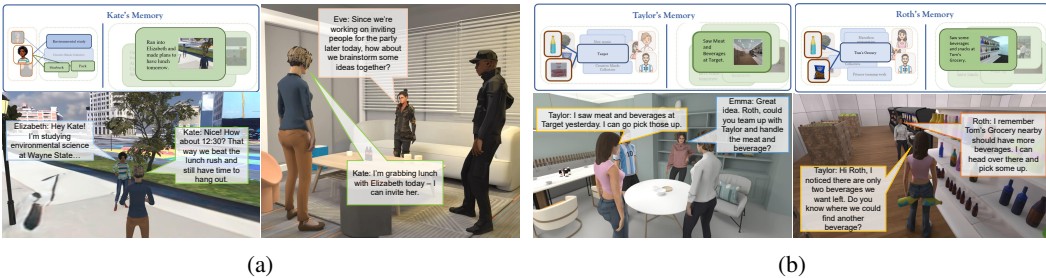

Figure 4: **Example behaviors demonstrating how *Ella* builds and leverages long-term memory.** (a) On the first day, Kate meets Elizabeth and establishes a connection, which later enables her to invite Elizabeth to their group's party during the Influence Battle. (b) Taylor and Roth draw on their knowledge of store locations and available supplies, allowing their group to make more efficient decisions in obtaining the target items.

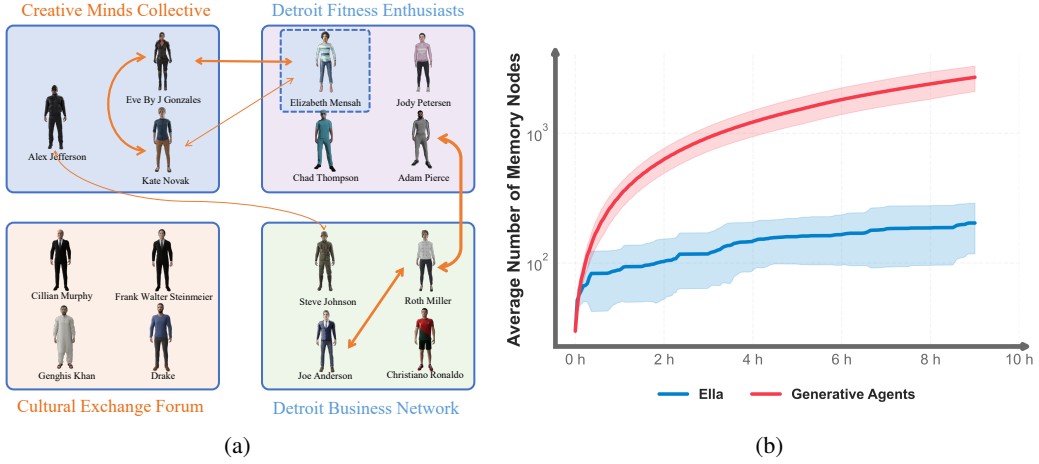

Figure 5: (a) **Social interaction pattern in *Influence Battle*.** The thickness of a line reflects the frequency of interaction. Members from Creative Minds Collective successfully persuaded Elizabeth Mensah to join their group's party. (b) **Comparison of memory growth over time.** The number of memory nodes averaged over 15 agents is shown here. Our structured memory system allows for more stable and organized growth.

`ViT-B-32-256` from openclip for images and `text-embedding-3-small` from Azure for text. We use `gpt-4o`[1] as the foundation model backbone for our method and *CoELA*, and `gpt-35-turbo`[2] for *Generative Agent*[3]. We also test our method with open source foundation models `DeepSeek-R1-Distill-Qwen-14B` and `Qwen2.5-14B-Instruct` served with vLLM (Kwon et al., 2023) in the experiments with oracle perception.

## 5.2 RESULTS

***Ella* can effectively accumulate experiences and acquire knowledge with the proposed long-term memory.** As shown in Figure 1b, *Ella* continuously accumulates new experiences and acquires new knowledge on the first day, covering nearly 50% of the environment. An example of the final spatial coverage in the Detroit community is illustrated in Figure 11 in the Appendix. Two example behaviors in Figure 4 show *Ella* effectively builds memory of other agents and the environment, which helps them make better decisions in the controlled evaluations.

***Ella*'s structured long-term memory is efficient.** Figure 5b further shows that *Ella*'s structured memory system allows for more stable and organized growth of memory nodes compared to the Generative Agents baseline. This structure enables more efficient retrieval as memory scales, supporting timely access to relevant events even as the memory grows.

***Ella* can influence other agents effectively.** As shown in Table 1, *Ella* achieves a higher show-up rate in the Influence Battle by successfully inviting more agents to the party across all three communities. This demonstrates its strong capabilities in social reasoning and persuasion. Although

---

[1] model version 2024-11-20

[2] model version 0125

[3] We tried to implement Generative Agent with `gpt-4o`, but the original prompts broke often and it's too costly given its large quantity of API call

| | Influence Battle | | Leadership Quest | |
|---|---|---|---|---|
| | show-up rate | # conv | completion rate | # conv |
| gpt-4o-1120 | 57.8 | 18.7 | 33.2 | 49.3 |
| **DeepSeek-R1-14B** | **40.0** | **48.3** | **8.0** | **46.0** |
| Qwen2.5-14B | 22.2 | 89.0 | 1.0 | 57.3 |

Table 2: **Results with open-source foundation model backbone.** We report the results of *Ella w/ Oracle Perception* with different backbones averaged over three communities.

| | Influence Battle | Leadership Quest |
|---|---|---|
| Full model (ours) | 46.7 | 37.5 |
| *+ importance* | 46.7 | 33.3 |
| *- spatial proximity* | 40.0 | 25.0 |
| *- multimodal data* | 33.3 | 33.3 |
| *- temporal recency* | 33.3 | 28.2 |

Table 3: **Ablation results** on the Detroit community. We report the main metric here.

the *CoELA* baseline engaged in twice as many conversations as *Ella*, its show-up rate was only half as high. This discrepancy arises from its lack of long-term memory, preventing it from effectively leveraging connections built the day before or recalling the party details after several hours (buried in new information from thousands of simulation steps). Meanwhile, Generative Agents engaged in so few conversations that they failed to invite other agents, despite being explicitly instructed to do so in their current community goals. As illustrated in Figure 5a, the party news propagates over time through the efforts of the organizer agents.

***Ella* can lead the group well.** As shown in Table 1, *Ella* completes four times more goals than other baselines in the Leadership Quest. Notably, *CoELA* had a completion rate of zero across all scenes—except in Detroit, where the leader partially completed the task alone—despite engaging in numerous conversations. This failure stems from its inability to retain the required items in memory. Among all scenarios, the London community posed the greatest challenge, where only *Ella* achieved a non-zero performance, demonstrating the robustness of our approach.

**Robust perception is important for embodied social agents.** Different from Park et al. (2023)'s setting where two agents knowing each other could only engage in a conversation when situated in the same grid, or Zhang et al. (2023)'s setting where two already-known agents could converse with each other anytime anywhere, our setting requires the agent to identify the agent to talk to according to their visual appearance or conversation contents and calculate the transmission range of their message according to the 3D location of the target agents to converse with, therefore a robust perception is critical for the agents to engage in social interactions in a 3D world. To isolate this perception challenge, we include an *Oracle Perception* variant. Comparing *Ella* with *w/ Oracle Perception* in Table 1, we observe meaningful performance gains: agents engage in more successful conversations and coordinated activities because they can more confidently and consistently identify one another and interpret the scene. This highlights that perception is a key limitation for embodied social agents, and motivates future work on improving robust 3D perception in open-world settings.

**Open source foundation models backbone is promising.** With advancements in open-source foundation models like DeepSeek-R1 (Guo et al., 2025), we wonder how well our framework works out-of-the-box on open-source foundation model backbones. We test *Ella w/ Oracle Perception* Agent with different backbones across all three communities and the two controlled evaluations, the results are shown in Table 2. Using DeepSeek-R1-Distill-Qwen-14B as the backbone without any further prompt engineering, *Ella w/ Oracle Perception* achieves a reasonable performance close to that of using a backbone of gpt-4o, while Qwen2.5-14B-Instruct performs much worse.

**Ablation study** To assess the contribution of each retrieval criterion, we perform an ablation study on four variants of our method in the Detroit community:

- add a criterion of *importance* during retrieval

- removes spatial information and the criterion of *spatial proximity*

- remove multimodal data in the episodic memory image by only calculating text embedding similarity for *content relevance* during retrieval

- remove *temporal recency* during retrieval

Table 3 reports the results. Across both tasks, performance drops consistently when any single criterion is removed, indicating that all three components, multimodal content relevance, spatial proximity, and temporal recency, contribute meaningfully and complement one another in supporting effective long-term memory retrieval.

## 6 LIMITATIONS

**Leverage the graph structure of the name-centric semantic memory.** Although the name-centric semantic memory is maintained as a graph structure, the current implementation retrieves knowledge based solely on text and image feature similarity. Enhancing our memory system with more sophisticated graph-based retrieval methods (Zhang et al., 2025; Sun et al., 2023; Gutiérrez et al., 2024) could enable effective multi-hop reasoning, paving the way for addressing reasoning-intensive challenges. This represents a promising direction for future work.

**All agents' thinking processes are assumed to finish synchronously.** Human cognition is bounded by limited computational resources (Lieder & Griffiths, 2020). In our current setting, Agents are assumed to *think* synchronously with unlimited computational resources, which means whatever deliberate the agent's thinking process is, it costs only 1 second in their world. It is interesting to examine the time cost of reasoning under explicitly limited computational resources, and to study how agents can adaptively switch between slow System 2 and fast System 1 thinking (Evans, 2003).

## 7 CONCLUSION

In this work, we build a non-parametric long-term memory for embodied agents with name-centric semantic memory and spatiotemporal episodic memory. We introduce *Ella*, an embodied social agent that uses foundation models and retrieved memory to reason, make daily plans, and engage in social activities. We conducted capability-oriented experiments in the Virtual Community with 15 agents in 3 different communities and demonstrated *Ella* can use long-term memory effectively to influence, cooperate, and lead other agents in an open world while accumulating multi-modal experience and acquiring knowledge continuously from visual observations of the environment and social interactions with other agents. Our findings imply the power of combining non-parametric long-term memory and foundation models to advance embodied general intelligence that could co-exist with humans.

ETHICS STATEMENT

As embodied social agents become more advanced, their integration into human-centered environments raises critical ethical and societal considerations. It's important to design and follow best practices in human-AI interactions (Amershi et al., 2019).

One key concern is the impact of AI-driven persuasion on human and agent interactions. In our ***Influence Battle*** evaluation, *Ella* successfully convinces other agents to attend an event, demonstrating its ability to shape group behavior. While such social reasoning capabilities are essential for cooperative AI, they could be misused in real-world applications, leading to manipulation, misinformation, or undue influence. To mitigate this, AI-driven persuasive agents must be designed with transparent intent disclosure and value alignment, ensuring they do not engage in deceptive or coercive behaviors.

Another concern is that their decision-making processes may inadvertently reflect and reinforce societal biases present in their training data or interaction patterns. For example, in our ***Leadership Quest***, Ella demonstrated superior leadership capabilities, but the fairness of leadership selection criteria in AI-driven systems remains an open question. Ensuring diversity and fairness in AI leadership roles requires robust bias mitigation strategies, careful dataset curation, and continuous evaluation of AI decision-making in diverse social contexts.

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

# A  ADDITIONAL EXPERIMENT DETAILS

## A.1  VIRTUAL COMMUNITY

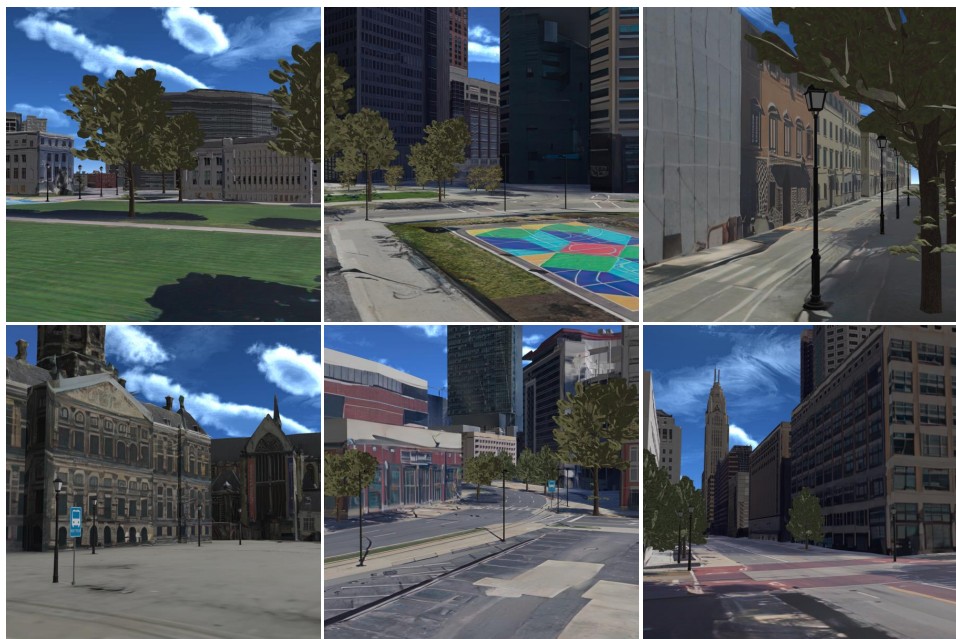

Figure 6: **Close-up views of different scenes in Virtual Community.**

Virtual Community (ViCo), introduced by Zhou et al. (2025), is an open world simulation platform for multi-agent embodied AI, featuring large-scale community scenarios derived from the real world with realistic physics and renderings. It was developed using Genesis (Authors, 2024) as its core engine, a generative physics simulator capable of modeling a wide variety of materials and an extensive array of robotic tasks, all while maintaining full differentiability. Additionally, Genesis features a real-time renderer based on OpenGL and a path-tracing renderer powered by Luisa. ViCo primarily offers scalable 3D scene creation and the generation of an embodied agent community.

ViCo develops an online pipeline to transform existing 3D geospatial data into high-quality simulation-ready scenes. Moreover, the pipeline automatically annotates the scenes from these geospatial data to facilitate real-world alignment. It supports the creation of expansive outdoor and indoor environments at any location and scale. Currently, ViCo has generated 57 scenes of various cities worldwide. In this paper, we use a subset of 3 scenes from the generated scenes for our evaluation: New York City, Detroit, and London. Figure 6 presents views of different scenes within Virtual Community.

ViCo has 74 avatar skins, consisting of skins retrieved from the Mixamo [4] and generated from real-world images using Avatar SDK [5]. We randomly sampled 15 skins for each of the 3 scenes. ViCo combines SMPL-X human skeletons (Pavlakos et al., 2019) with created avatar skins to support up to 2,299 unique motions from Mixamo. Additionally, ViCo can generate scene-grounded characters that are socially connected at a community level. Figure 7 illustrates a generated community in New York City with places of different functionalities annotated.

## A.2  TASK DETAILS

To evaluate the effectiveness of the proposed non-parametric memory and high-level cognitive capabilities of the embodied agents, we design our experiments in two stages as shown in Figure 8. In the first stage, 15 agents are simulated for 9 hours (34200 steps) for their first day in the community, during which the agents could familiarize themselves with the 600m * 600m scene and other agents

---

[4] https://www.mixamo.com/
[5] https://avatarsdk.com

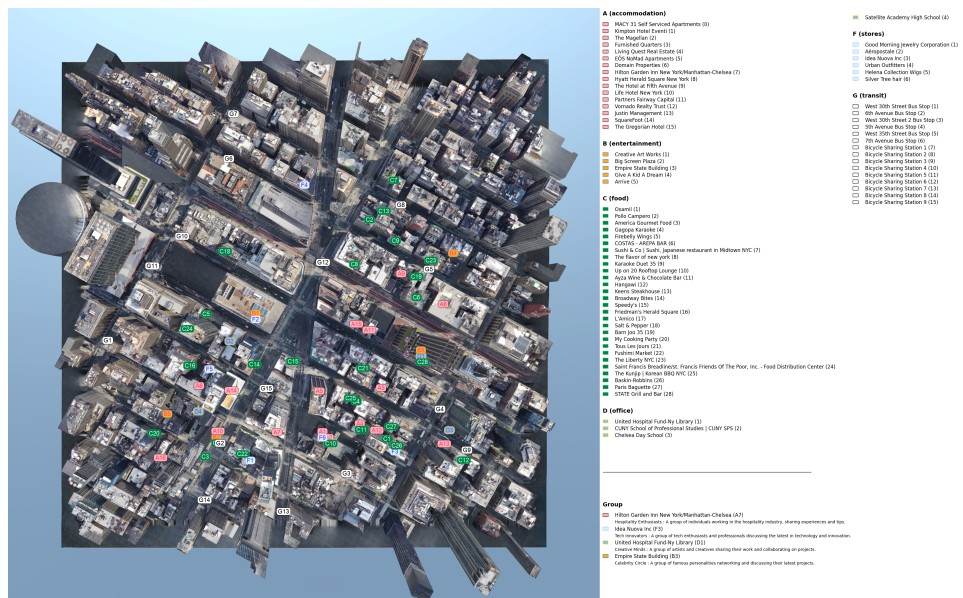

Figure 7: **An illustration of a community in New York City with places of different functionalities annotated.** There are 6 types of functional places: accommodation, entertainment, food, office, stores, and transit, each labeled with different colors on the figure. Social group information is also annotated with the group name, the group meeting place, and the group description.

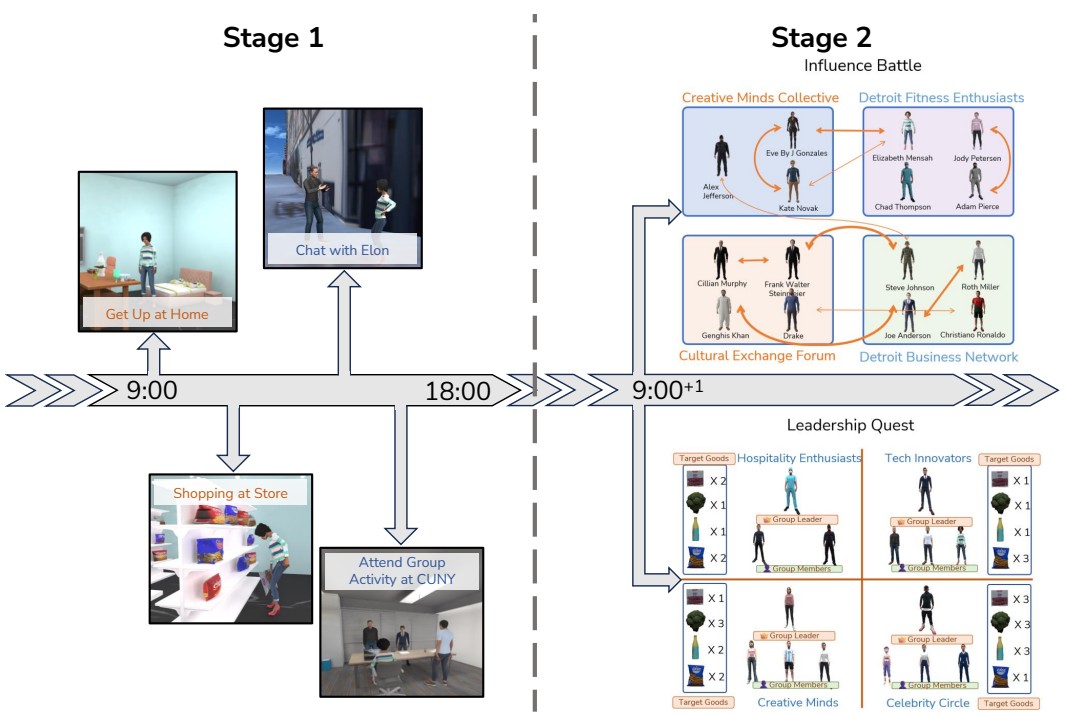

Figure 8: **Two stages of the experiments.** The embodied agents first live a 9-hour social life with diverse community goals in stage 1, then are evaluated with two controlled evaluations in stage 2.

and build memories. Then in the second stage, we test them with two controlled evaluations in the days following: *Influence Battle* and *Leadership Quest*. We select these tasks because they probe meaningful social-cognitive capabilities that require long-term memory, social reasoning, and spatial grounding—rather than narrowly defined short-term behaviors. *Influence Battle* is inspired by

> **Community Goal Prompt:**
>
> My group $group_name$ is organizing a party at $group_place$ from 14:30:00 to 15:00:00 today. I need to go around the city, find and invite people outside of my group to attend our party today.

Figure 9: **Prompt for assigning community goals in *Influence Battle*.** `$group_name$` is replaced with the agent's group name, `$group_place$` is replaced with the agent's group place.

> **Community Goal Prompt for Group Leaders:**
>
> I am the leader of my group $group_name$. I need to discuss and assign tasks to my group members to collect $target_items$ from stores and bring them back here at $group_place$ before 12:00:00. Each person could take 2 items at a time with each of his hand.
>
> **Community Goal Prompt for Group Members:**
>
> I need to help my group $group_name$'s leader $leader$ to prepare for a group activity. I will discuss with my leader about the items to collect, follow the instructions given by my leader and complete my assigned task before 12:00:00.

Figure 10: **Prompt for assigning community goals in *Leadership Quest*.** `$group_name$` is replaced with the agent's group name, `$group_place$` is replaced with the agent's group place, `$target_items$` is replaced with a string containing all target items and the quantity, `$leader$` is replaced with the name of the leader of the group.

the line of research on multi-agent influence and persuasion in mixed-motive environments (Jaques et al., 2019; Falk & Scholz, 2018), ***Leadership Quest*** builds on longstanding work in multi-agent coordination and leadership assignment (Srivastava et al., 2006; Guo et al., 2024). In ***Influence Battle***, two of the four groups will be asked to organize a party at a specific place in 6 hours, and the members need to go around the city, find, and invite agents outside of their group to attend the party. This evaluation tests the agents' capability to impact other agents by persuading them to attend the parties, which requires the capability of social reasoning, persuasion, and decision-making. In ***Leadership Quest***, each of the four groups is assigned a task to purchase several items from various stores in the city and return within 3 hours. One member from each group is designated as the leader and is the only one given full details of the task, while the remaining members are simply instructed to assist the leader. This controlled evaluation setting challenges the agent's leadership abilities, particularly in assigning sub-tasks based on the diverse personalities and resources of group members.

The observation includes posed $512 \times 512$ RGB and depth images, the content of the heard messages within range, and current states including pose, place, time, cash, held objects, and vehicles being taken. The agent's action space consists of navigation actions of *move forward $x$ m*, *turn left $x$ degree*, *turn right $x$ degree*, *enter $x$ place or vehicle*, and *exit $x$ vehicle*; interaction actions of *pick $x$ object with hand $y$*, *drop object in hand $x$*; and *converse message $x$ with a range of $y$ m*. The message transmission range threshold $\theta_{msg}$ is set to 10m.

We include the detailed prompt used to assign the community goal to agents in two tasks in Figure 9 and Figure 10.

### A.3 COMPUTE

We conducted our experiments using a single NVIDIA A100 GPU. Stage one of each community life simulation was run for 20 hours, while stage two of each task and community was executed for an additional 10 hours. On average, each agent's saved memory—including both episodic and semantic components—occupies approximately 161 MB after 9 hours of simulation. During runtime, agents consume additional memory for perception, planning, and retrieval. In particular, the perception module alone requires around 4 GB of GPU memory per agent. The peak RAM usage per agent process is approximately 1 GB.

**Lifelong simulation of a community of agents in a visually rich, physics-realistic environment is computationally expensive.** Although our experiments span only 1.5 simulated days—seemingly short for a "lifelong" setting—we adopt the widely used interpretation of lifelong learning as an agent's ability to accumulate, retain, and reuse knowledge across experiences (Chen & Liu, 2018). Despite extensive system-level optimizations to accelerate simulation, each simulated second still requires at least one second of real time. This is due to the intensive demands of multi-camera rendering, skinned motion computation, and the invocation of multiple models or APIs during each agent's decision-making process. As a result, simulating one day in the environment consumes an entire real-world day, significantly constraining the scale of experimentation. Continued progress in graphics and simulation technologies is expected to ease this bottleneck and support faster development of embodied social agents in high-fidelity, physics-grounded environments.

## A.4 Additional Discussions

**Perception error propagation and mitigation** Perception errors inevitably occur in an open-world 3D environment and can propagate into an agent's long-term memory, influencing later retrieval, planning, and social interaction. In our current implementation, both episodic and semantic memories are updated directly from each new observation or conversation, without explicit confidence modeling or cross-observation consistency checks. As a result, misidentifications, such as confusing two agents, misclassifying an object, or localizing an entity imprecisely, may enter episodic memory and occasionally lead to suboptimal actions or incorrect assumptions during later queries. However, the system can tolerate a moderate level of such noise without causing uncontrolled cascading failures due to three mitigating factors. First, the retrieval module ranks memory items using joint multimodal relevance, spatial proximity, and temporal recency, which naturally downweights isolated noisy entries that do not consistently match a query along these dimensions. Second, conversational interactions often act as a verbal correction channel: misidentified agents in visual observations may later be clarified when their names or roles are explicitly referenced in dialogue, allowing the semantic memory to store accurate facts even when earlier episodic traces were noisy. Third, the dual-memory structure separates low-level event logs from higher-level semantic facts, so noisy episodic observations do not automatically overwrite established semantic knowledge. Together, these properties allow the memory system to resist some level of perception error without catastrophic drift, while developing stronger error-handling mechanisms remains a valuable direction for future research.

**Selective forgetting for lifelong operation** As the duration of an agent's life increases, long-term memory inevitably accumulates a large number of events, observations, and social interactions. While our current system does not implement an explicit forgetting mechanism, the hierarchical structure of the memory already provides a natural foundation for selective pruning at different levels of abstraction. Low-level spatial voxel information tends to be the most redundant and scene-specific. These low-level details can be pruned or downsampled over time without significantly affecting the integrity of the semantic memory. Similarly, episodic entries that are repeatedly superseded by stable semantic facts, such as a person's identity, home location, or profession, could be removed entirely when memory pressure increases. Such mechanisms would allow the agent to retain core long-term knowledge while discarding outdated or low-utility details, mirroring human forgetting processes that preserve essential semantic structure while compressing or discarding past experiences. Designing principled, query-aware forgetting strategies that balance stability and plasticity represents an important direction for enabling lifelong operation in even larger or unbounded environments.

**Clarification on naming.** To avoid confusion with prior work, we note that our embodied agent ELLA is unrelated to *ELLA: Efficient Lifelong Learning Algorithm* (Ruvolo & Eaton, 2013). The classical ELLA focuses on efficient parametric transfer across a sequence of supervised tasks, whereas our ELLA tackles embodied lifelong learning in an open 3D social world using non-parametric multimodal memory, long-horizon visual and social experience accumulation, and spatially grounded interaction. Our naming choice stems from the semantic meaning of "Ella" ("her" in Spanish), reflecting our long-term vision for socially and visually grounded embodied agents. We include this clarification to ensure readers do not conflate the two distinct lines of work.

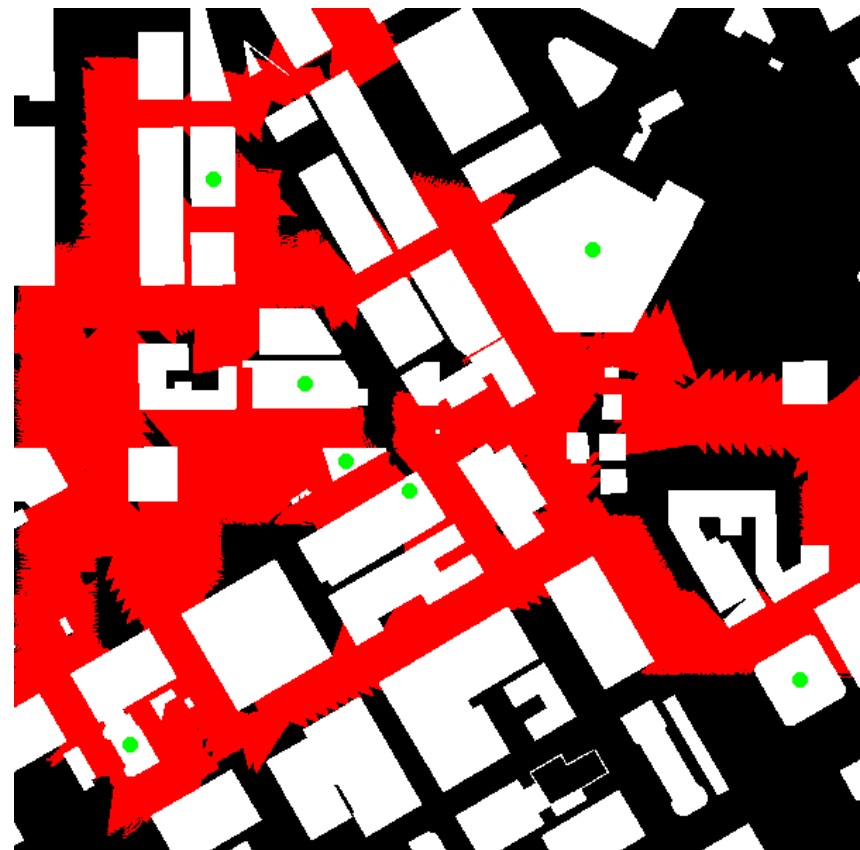

Figure 11: **A visualization of the final spatial coverage on the Detroit community.** Explored regions are shown in red, buildings are shown in white, and unexplored regions are shown in black. The buildings in the agent's schedule are denoted with green circles.

## B ADDITIONAL IMPLEMENTATION DETAILS

### B.1 NAVIGATION

Given the volume grid maintained in the semantic memory introduced in Section 4.1.1, we construct the occupancy map and partition the entire map into three types of grid points: unknown, known obstacles, and known non-obstacles, as illustrated in Figure 11. The A* algorithm is employed to search for the shortest path, where the weight of known non-obstacle points is set to 1, unknown points are assigned a weight of 5, and obstacle points are given an infinite weight. Additionally, to mitigate the issue of agents getting stuck near obstacles due to potential wall-clipping, points closer to obstacles are assigned higher weights. Specifically, a point at a distance $d$ from an obstacle is assigned an additional weight of $\frac{100}{d}$. Finally, to prevent the agent from wandering in place due to significant discrepancies between consecutive navigation paths, the previously computed path is prioritized unless it is found to be infeasible (i.e., it crosses an obstacle).

### B.2 COMMUTE

When an agent executes a *commute* activity, it must determine how to travel between locations using the available transit options in the community. *Ella* begins by retrieving knowledge about the transit system together with the spatial information of the origin and destination. This information is provided to a foundation model, which produces a *commute plan* composed of a sequence of travel segments (e.g., walking, bicycling, or taking a bus). Each segment in the generated plan is then executed using the navigation submodule described in Appendix B.1.

## B.3 BEHAVIOR PLANNING

For activities such as *meal* or *main*, the agent generates a structured behavior plan that specifies navigation steps and task-relevant motions. Upon starting the activity, *Ella* retrieves information about the objects and affordances available at the current location. This contextual information, together with the activity description and the agent's own profile, is passed to a foundation model to produce a *motion schedule*, which the agent then executes step-by-step.

## C PROMPT TEMPLATES

We provide the full prompt template for the modules introduced in Section 4.3 in Figure 12 - Figure 16.

---

**Prompt:**

Given my character description and retrieved memory, please help me plan tomorrow's schedule.

My Character Description:

$Character$

Current Situation:

$Context$

Schedule format: The output should be a JSON object which is an array of activities for the character. Each activity should follow the following format:

```
{
    "type": "activity type, should be one of the following: 'commute', 'meal', 'sleep', 'main'",
    "activity": "activity description",
    "place": "name of the place where the activity takes place, should be in the list of the known places. Should be null for commute activities",
    "building": "name of the building the activity place belongs to, should be consistent as in the list of known places. Should be null for commute activities",
    "start_time": "HH:MM:SS",
    "end_time": "HH:MM:SS",
}
```

Note: The schedule should be planned based on the character's description and known places. The place should be mentioned for each activity and must be included in the known places. Do not hallucinate places. Commute activities should be given enough time to finish and be inserted between all consecutive activities that do not share the same building so the agent can have time to commute to the correct building before the start of the activity, including commute to meal places. The schedule should start at 00:00:00 and end at 23:59:59, and covering the consecutive time of 24 hours with no gaps. The schedule should not end with a commute activity or an activity lasting over 23:59:59, so the character is not commuting when the day is ending. The schedule should be planned in a way that the character can complete all the activities within the given time frame.

Tomorrow is $Date$. My full schedule for tomorrow:

---

Figure 12: **Prompt template for generating the daily schedule.** `$Character$` is replaced with the agent's character description, `$Context$` is replaced with the retrieved memory.

Prompt:

Given my character description, current schedule and situation, help me determine what I should do.

My Character Description:

$Character$

My current schedule:

$Schedule$

Current Time:

$Time$

Current Place:

$Place$

Related Experiences:

$Experiences$

Current Situation:

$Context$

Note: There are four types of options: revise the schedule, interact with the environment, engage in a conversation, or continue doing current activity. Please help me choose the best option based on my situation. Output a JSON object with the following format:

```
{
   "option": "name of the option",
   "target": "For 'engage in a conversation,' include the person's full name if known. For 'interaction with the environment,' include the action name and object name. Otherwise, set to null.",
   "reason": "Explain why this option is the best choice given the context."
}
```

Figure 13: **Prompt template for generating the reaction.** $Character$ is replaced with the agent's character description, $Schedule$ is replaced with today's remaining schedules, $Experience$ is replaced with the retrieved memory, $Context$ is replaced with the latest memory.

Prompt:

Given my character description, knowledge about and experience with $Target_name$, and current situation, help me decide what I should say next.

My Character Description:

$Character$

My knowledge about $Target_name$:

$Target_knowledge$

My experience with $Target_name$:

$Target_experience$

Current Place:

$Place$

Current Time:

$Time$

Current Situation:

$Context$

Current conversation history:

$Conversation_history$

Note: Please generate a short utterance of what I should say next to $Target_name$, which should be null if the conversation should be ended now. Output a JSON object with the following format:

```
{
   "utterance": "short utterance of what I should say next to $Target_name$, null if the conversation should be ended now.",
   "reason": "reason for generating this utterance"
}
```

Figure 14: **Prompt template for generating the utterance.** `$Character$` is replaced with the agent's character description, `$Target_knowledge$`, `$Target_experience$`, `$Context$` are replaced with the retrieved memory, `$Conversation_history$` is replaced with the last 4 messages.

Prompt:

I just had a conversation with $Target_name$. Summarize it in one sentence.

Full conversation:

$Conversation_history$

Note: Output only the summary of the conversation in one sentence. Do not include any other information.

Figure 15: **Prompt template for generating the summarization of the conversation.** `$Conversation_history$` is replaced with the full conversation.

```
Prompt:

I just had a conversation with $Target_name$. Help me extract new knowledge I learned from it.

Full conversation:

$Conversation_history$

Note: Output a JSON object which is an array of knowledge items. Each knowledge item should follow the
following format:

{
   "name": "name of the knowledge item",
   "description": "description of the knowledge item"
   other fields
}

Example knowledge items I already have:

$Knowledge_items$

New knowledge items I learned from the conversation:
```

Figure 16: **Prompt template for extracting knowledge from a conversation.** $Conversation\_history$ is replaced with the full conversation, $Knowledge\_items$ is replaced with sampled knowledge items from semantic memory.

