# OpenReview forum: "Ella: Embodied Lifelong Learning Agents with Non-Parametric Memory"
_ICLR.cc/2026/Conference — Submitted to ICLR 2026_

### Official Review · Reviewer_k6G2 · 2025-10-31

**Soundness:** 1
**Presentation:** 3
**Contribution:** 1
**Rating:** 2
**Confidence:** 3

**Summary:**

ELLA is a system for an autonomous agent designed to operate on the order of days. It combines a long-term spatial memory based on multi-layer 3D scene graphs, along with LLM and multimodal models.

The authors demonstrate a case study in a multi-agent by running several copies agents in a model scenario where agents in a town are asked to plan a party. The authors give somewhat elaborate personalities to each of the agents, and designate 4 predetermined social groups such as "fitness enthusiast" and "ai enthusiast".

**Strengths:**

The paper tackles an ambitious and important problem - enabling embodied agents to learn and operate over extended temporal scales (days) in social environments. I found it a fun read, and there were plenty of good cogsci + psychology references.
* The evaluation setup is creative and goes beyond typical navigation or manipulation tasks.
* The paper includes ablation studies with oracle perception, showing the impact of perception errors on the overall system performance.
* The inclusion of experiments with open-source foundation models (DeepSeek-R1, Qwen2.5) shows the system depends heavily on the LLM used

**Weaknesses:**

Overall this is an interesting setup, but the paper introduces several things, and currently the experiments do not feel complete or well-matched to the specific things introduced.

**Benchmark**
On the one hand, the paper introduces a multi-agent setting and setup. This is the only evaluation used; a multi-agent setup on two hand-defined tasks in a single hand-defined setting. If this is the main focus of the submission, then considering progress in the field, I think there must be more recent baselines that have come out in the past two years and could be compared in the benchmark? Why were these tasks and metrrics chosen -- what behaviors do they measure and why are existing benchmarks not sufficient. Multi-agent is not my main familiarity, so I leave this to the other reviewers.

**Memory**
On the other hand, this paper introduces a RAG-like memory that queries a 3D scene graph and combines it with an LLM. It's not clear to me if this is supposed to be a core contribution of the paper; it is listed in the contributions -- but seems to play a minor role if the focus is on the multi-agent setting where oracle perception could be used.

If the scene graph is important, then why is the multi-agent setting needed to evaluate the efficacy of the scene graph, and why do existing long-term memories not suffice and where does this plug the gap? I see that this plugs a gap with the Coela baeline, but other existing work can, too.

**4.1.1: Hierarchical scene graph as spatial memory**
There are plenty of 3D scene graph approaches that could serve as a starting point and are intended to support long-term memory that interfaces nicely with LLMs, with established benchmarks for single-agent performance (e.g. SayPlan https://sayplan.github.io/, or something like this: https://arxiv.org/pdf/2510.16643 -- the second paper came our recently; I just found it by googling. The point is this _type_ of eval, not necessarily a comparison to this specific paper.)

The authors do cite a couple scene graph works (e.g. conceptgraphs, hydra), but the actual method is not explicitly related to any of them. My opinion is that submission would be stronger if the method itself built more off of existing literature -- it would be easier to pinpoint the contribution of this paper, and fewer ablations would be needed to understand the design choices.

For the specific long-term memory module; there are no ablations, except maybe figure 1b where the analysis is simply how many nodes stored (not whether it is effective, or how effective compared to baselines.


**4.2 Spatiotemporal Episodic Memory**
The authors propose a handcrafted retrieval rule, similar to conceptgraphs, that builds on feature similarity and spatiotemporal recency. There are no ablations or analysis of the design choices here.

**Questions:**

There is a classical lifelong learning paper "ELLA: Efficient Lifelong Learning Algorithm" https://proceedings.mlr.press/v28/ruvolo13.html that may be confusing for some readers given the name. I just wanted to bring this to the authors' attention -- perhaps they could consider some ways to avoid confusion. Modifying the acronym, letting the reader know, etc.

Benchmark:
* How were the personalities for the characters in the scenario chosen?
* Do all agents in the scenario in all groups use the same agent (e.g. ELLA or Coela or Generative Agents)?

Memory:
* I have some questions about the generality of the approach. Since the model builds in a layer for buildings, would it work outdoors, in a forest, or underwater?

---

> ### Author Response · Authors · 2025-11-24
> **Response to Reviewer k6G2 [1/3]**
>
> *We sincerely thank the reviewer for the time reading our paper, and we appreciate your insightful and constructive comments! We address your concerns in detail below and have updated our paper according to your suggestions.*
>
> > Q1: Is the memory and integration with llm a core contribution of the paper? it is listed in the contributions -- but seems to play a minor role if the focus is on the multi-agent setting where oracle perception could be used. If the scene graph is important, then why is the multi-agent setting needed to evaluate the efficacy of the scene graph, and why do existing long-term memories not suffice and where does this plug the gap? I see that this plugs a gap with the Coela baeline, but other existing work can, too.
>
> Yes, our core contribution is the **dual-form long-term memory**, and its **integration with foudation models for lifelong embodied decision making**. Our setting is fundamentally different from (a) a multi-agent setting with oracle perception and (b) a single-agent embodied perception setting.
>
> Our problem formulation targets **embodied lifelong learning in an open, social 3D world**, where agents accumulate knowledge from **both** **visual perception** and **social interactions**. This dual modality of experience is essential, humans continuously learn from visual perceptions and social interactions, and existing long-term memory systems do not support this unified learning process. To the best of our knowledge, we're the first to build such memory system that can learn from both visual observations and social interactions. If there are prior works achieving this unified memory across visual and social experiences, we would be grateful for the pointers and happy to include direct comparisons.
>
> > Q2: Is the Benchmark the main focus of the paper? If so, I think there must be more recent baselines that have come out in the past two years and could be compared in the benchmark?
>
> Benchmarking is **not** the main focus. The benchmark is introduced because **no existing evaluation setting** covers our problem: embodied lifelong learning across visual perceptions and social interactions in an open community of agents.
>
> Thus, we adapted the two closest baselines that can operate in our setting. If the reviewer is aware of additional methods that can (i) handle multi-agent social interactions, (ii) learn from long-horizon visual experience, and (iii) operate in a large open 3D world, we would be happy to incorporate them.
>
> > Q3: Why were these tasks and metrrics chosen -- what behaviors do they measure and why are existing benchmarks not sufficient.
>
> Thanks for the valuable question. As discussed in Lines 91–92, our goal is to evaluate **higher-level cognitive capabilities in a lifelong setting**, consistent with the vision outlined in [1]. We therefore design two **capability-oriented** evaluations after a full day of unsupervised experience:
>
> - Influence Battle: measures social reasoning, persuasion, and long-horizon planning to influence other agents' decisions.
> - Leadership Quest: measures the ability to coordinate diverse agents with different personalities and resources.
>
> We select these tasks because they probe meaningful social-cognitive capabilities that require long-term memory, social reasoning, and spatial grounding—rather than narrowly defined short-term behaviors. *Influence Battle* is inspired by the line of research on multi-agent influence and persuasion in mixed-motive environments[2][3], *Leadership Quest* builds on longstanding work in multi-agent coordination and leadership assignment [4][5].
>
> These behaviors are extremely challenging to quantify directly. Therefore, we adopt concrete, easily computed metrics, *show-up rate* and *average completion rate*, which correlate closely with the high-level capabilities we are measuring. To the best of our knowledge, there is no other existing benchmarks sharing our setting of embodied lieflong learning in an **open and social** world, which requires open 3D environments and community of socially-connected embodied agents. Therefore, existing single-agent benchmarks cannot evaluate the core capabilities we target.

---

> ### Author Response · Authors · 2025-11-24
> **Response to Reviewer k6G2 [2/3]**
>
> >Q4: There are plenty of 3D scene graph approaches that could serve as a starting point and are intended to support long-term memory that interfaces nicely with LLMs, with established benchmarks for single-agent performance (e.g. SayPlan https://sayplan.github.io/, or something like this: https://arxiv.org/pdf/2510.16643 -- the second paper came our recently; I just found it by googling. The point is this type of eval, not necessarily a comparison to this specific paper.)
>
> We appreciate the helpful references and have included them. These works are highly valuable, but they evaluate short-horizon, single-agent tasks. Our focus is fundamentally different: building long-horizon memory from both visual observations and social interactions across many agents in a community-scale environment. As stated in Related Work 2.2, these single-agent benchmarks do not capture the challenges—persistent social memory, multi-agent interactions, and large-scale environments—that our work studies.
>
>
> > Q5: The authors do cite a couple scene graph works (e.g. conceptgraphs, hydra), but the actual method is not explicitly related to any of them. My opinion is that submission would be stronger if the method itself built more off of existing literature -- it would be easier to pinpoint the contribution of this paper, and fewer ablations would be needed to understand the design choices.
>
> Thank you for the suggestion to strength our work! To clarify, our method directly builds on this literature. As clarified in Lines 201, 210, and 251, our multi-stage perception pipeline follows the same principles as prior 3D scene-graph systems, and our region-layer construction using GVD is inspired by Hydra. We have strengthened these connections in the revision to make the relationships more explicit.
>
> > Q6: For the specific long-term memory module; there are no ablations, except maybe figure 1b where the analysis is simply how many nodes stored (not whether it is effective, or how effective compared to baselines.
>
> We apologize for the confusion: Figure 1b is **not** intended as an ablation but simply illustrates memory growth over time. We **do evaluate the effectiveness of our memory module**. Figure 5(b) and Lines 419–422 compare our dual-form memory to the baseline and show its clear efficiency.
>
>
> > Q7:  The authors propose a handcrafted retrieval rule, similar to conceptgraphs, that builds on feature similarity and spatiotemporal recency. There are no ablations or analysis of the design choices here.
>
> Thank you for raising this. Our retrieval is more refined than the two strategies tested in ConceptGraph. We use three complementary criteria:
>
> - Content relevance (CLIP similarity + text-embedding similarity)
> - Temporal recency
> - Spatial proximity
>
> Unlike ConceptGraph, our method *does not* enumerate all objects in the LLM context—this is infeasible in our large open-world setting.
>
> We have added new ablation results (now in Table.3 and section 5.2) showing that **each component contributes meaningfully** to performance.
>
> | Setting                | Influence Battle | Leadership Quest |
> |------------------------|------------------|------------------|
> | Full model (ours)      | 46.7             | 37.5           |
> | + *importance*         | 46.7             | 33.3           |
> | - *spatial proximity*  | 40.0             | 25.0           |
> | - *multimodal data*    | 33.3             | 33.3           |
> | - *temporal recency*   | 33.3             | 28.2           |

---

> ### Author Response · Authors · 2025-11-24
> **Response to Reviewer k6G2 [3/3]**
>
> > Q8: There is a classical lifelong learning paper "ELLA: Efficient Lifelong Learning Algorithm" https://proceedings.mlr.press/v28/ruvolo13.html that may be confusing for some readers given the name.
>
> We appreciate the pointer. We acknowledge the name overlap and now clarify in Appendix.A.4 that our “ELLA” differs entirely in motivation and functionality. We chose the name for its semantic meaning (“her” in Spanish), reflecting our long-term vision for embodied agents learning socially and visually.
>
> > Q9: How were the personalities for the characters in the scenario chosen?
>
> All personalities are generated by Virtual Community conditioned on the specific scene. This ensures scene grounding, internal social consistency, and diversity across groups.
>
>
> > Q10: Do all agents in the scenario in all groups use the same agent (e.g. ELLA or Coela or Generative Agents)?
>
> Yes. All agents use the same underlying agent architecture, ensuring fair comparison in competetive multi-agent evaluation.
>
> > Q11: Since the model builds in a layer for buildings, would it work outdoors, in a forest, or underwater?
>
> Thanks for the insightful question. Our region layer adopts the concept of “buildings” as an inductive bias because our primary experiments focus on city environments where humans build modern social relationships. In forest or underwater scenes, region layer could be constructed with other inductive bias like the plants distribution. The memory architecture and pipeline remain applicable.
>
> *[1] The animal-ai olympics. Nature Machine Intelligence 2019*
>
> *[2] Social Influence as Intrinsic Motivation for Multi-Agent Deep Reinforcement Learning. ICML 2019*
>
> *[3] Persuasion, influence, and value: Perspectives from communication and social neuroscience. Annual Review of Psychology 2018*
>
> *[4] Empowering leadership in management teams: effects on knowledge sharing, efficacy, and performance. Academy of Management Journal 2006*
>
> *[5] Embodied LLM Agents Learn to Cooperate in Organized Teams. NeurIPS 2024*
>
> *We sincerely appreciate your constructive suggestions, and hope our new experiments and clarifications have addressed your concerns and turned your assesment positive. Please feel free to let us know if you have further questions.*
>
>
> Best,
>
> Authors

---

### Official Review · Reviewer_j9Ji · 2025-11-01

**Soundness:** 3
**Presentation:** 3
**Contribution:** 3
**Rating:** 6
**Confidence:** 3

**Summary:**

This paper introduces Ella, an embodied agent designed for lifelong learning within a social, 3D open-world environment. The central challenge the paper addresses is the inability of current foundation model-based agents to continuously learn from new information over long periods, often suffering from catastrophic forgetting when parameters are fine-tuned. The core contribution is Ella's structured, non-parametric, long-term multi-modal memory system. Ella integrates this memory system with foundation models using a planning-reaction framework.

**Strengths:**

1. This paper introduce a novel and grounded memory architecture which utilizes dual semantic/episodic memory.
2. Ella demonstrates a clear and significant performance advantage over baselines in the new social tasks (Table 1). The analysis effectively pinpoints the baselines' failures.
3. This paper also includes thoughtful analysis which is a good base for future work.

**Weaknesses:**

1. As the authors note, the retrieval system is based on feature similarity and does not fully exploit the rich graph structure of the name-centric semantic memory. This limits the agent's reasoning to relatively simple, direct lookups rather than complex, multi-hop queries (e.g., "Who did I meet at the place where I bought coffee yesterday?").
2.  The large performance gap between Ella and "Ella + Oracle Perception" suggests that the current perception pipeline is a major bottleneck. While using open-set models is a strength, the paper does not deeply analyze how perception errors (e.g., misidentifying agents, failing to track objects) propagate into the long-term memory and cause cascading failures in reasoning.

**Questions:**

1. How does memory scales? The 9-hour simulation generated ~161MB of memory per agent. If you were to run this simulation for a simulated month, what new challenges would you anticipate for the retrieval system?

---

> ### Author Response · Authors · 2025-11-24
> **Response to Reviewer j9Ji**
>
> *We sincerely thank the reviewer for the time to read our paper, and we appreciate your positive and constructive feedback! We address your questions in detail below and have updated our paper according to your suggestions.*
>
> > Q1: As the authors note, the retrieval system is based on feature similarity and does not fully exploit the rich graph structure of the name-centric semantic memory. This limits the agent's reasoning to relatively simple, direct lookups rather than complex, multi-hop queries (e.g., "Who did I meet at the place where I bought coffee yesterday?").
>
> Thank you for this insightful observation. We completely agree that leveraging the full structure of the name-centric semantic memory, beyond similarity-based retrieval, would enable richer multi-step reasoning capabilities. Incorporating graph-based retrieval and multi-hop reasoning methods, such as Think-on-Graph [1] and HippoRAG [2], is an exciting direction for future work and aligns closely with the long-term goals of our system.
>
> > Q2: The paper does not deeply analyze how perception errors (e.g., misidentifying agents, failing to track objects) propagate into the long-term memory and cause cascading failures in reasoning.
>
> Thank you for highlighting this important point. Following your suggestion, we **added an analysis of perception-error propagation** in the revised paper. We discuss how misdetections and identity confusion enter the episodic memory, how they can affect subsequent retrieval and planning, and how our structured memory design mitigates some of these cascading effects. We believe this additional analysis strengthens the clarity of the system’s limitations and future opportunities.
>
> > Q3: How does memory scales? The 9-hour simulation generated ~161MB of memory per agent. If you were to run this simulation for a simulated month, what new challenges would you anticipate for the retrieval system?
>
> We appreciate the thoughtful question. As discussed in section 5.2, Figure 5(b) shows our structured memory system enables more stable and organized growth compared to baselines. Over longer horizons (e.g., a simulated month), we expect:
>
> - Storage growth to slow down once the agent has explored most of the fixed 600m * 600m environment (in 9 hours it already covers ~50%).
> - Most new memory to arise from novel social interactions.
> - Our hierarchical structure to support selective pruning of low-level spatial details when memory grows too large, allowing scalable operation even in larger or unbounded environments.
>
> We have added this discussion to clarify the scalability behavior and future design considerations.
>
> *[1] Think-on-Graph: Deep and Responsible Reasoning of Large Language Model on Knowledge Graph. ICLR 2024*
>
> *[2] HippoRAG: Neurobiologically Inspired Long-Term Memory for Large Language Models. NeurIPS 2024*
>
> *We sincerely appreciate your comments. Please feel free to let us know if you have further questions.*
>
> Best,
>
> Authors

---

### Official Review · Reviewer_MVQs · 2025-11-03

**Soundness:** 2
**Presentation:** 1
**Contribution:** 1
**Rating:** 2
**Confidence:** 3

**Summary:**

This paper focuses on life long learning agents which act in situated environments. The agents have a long term spatial memory where they store precise spatial representations for locations and events that happen over time. These memories are used to plan, react and communicate with other agents in a continual learning setting. Summarized conversations are also stored in episodic memory which builds up over time. Experiments are performed in the “Virtual Community” environment with 15 unique agents.

**Strengths:**

The paper aims to tackle a very ambitious problem of learning in situated agents with non-parametric memory, where the agent environment simulation happens over a timescale of days. This is also a very important and challenging problem given that the current set of foundation models (and thus agents) have no explicit sense of memory and most often memory is added on top of these agents.

**Weaknesses:**

The paper is not really well written, it is very confusing in parts. It is not really well motivated at all. While I agree that agents need memory and some form of non-parameteric memory would be important, the paper does not present convincing arguments/experiments on why the agent would need precise spatial memory for high level semantic goals. The details on how the agent acts to generate low level actions is missing (Are they even generated, e.g. when the agent generates a goal “go to office”, how is this implemented?) All of these details are critical and completely missing from the paper. The experimental section is also quite week, there aren’t any ablations. Some of the experiment settings e.g. (need of robust perception for embodied social agents) are not motivated at all, instead results are presented directly. All of this makes it very hard to understand how the paper is implemented.

**Questions:**

*Hierarchical Scene Graph as Spatial Memory*: The paper goes into details on how the spatial memory is being created but it is not clear how it is being used by the agent. The paper mentions that the agent uses/needs spatial memory to act, but what is the action here? Are these navigation actions, manipulation actions or other high level semantic actions. If the latter then why do we need spatial memory. Also, it is quite unclear why precise spatial memory (e.g. which object is at what location) is needed in the future. If an agent goes from place A to place B, then the spatial locations of object in place A (spatial memory of A) is not really needed at place B.

*Daily Schedule*: The daily schedule of the agent shown in Figure 3 (c) consists of mostly high level goals (e.g. commute to office). How are these high level goals grounded in the environment? It seems relying on memory alone would be insufficient for many problems (e.g. navigation) which would require planning based on current context instead of past context?

The paper has a very very ambitious goal, to achieve memory for really really long horizon actions (e.g. at the level of days) when the environment clock is ticking at 1 FPS. Does this mean that the number of actions the agent takes in the environment is 1.5 days * 86400? Or does the paper assume some hierarchical actions. I think many such critical details are completely missing from the paper.

---

> ### Author Response · Authors · 2025-11-24
> **Response to Reviewer MVQs [1/2]**
>
> *We sincerely thank the reviewer for the time reading our paper, and we appreciate your thoughtful and constructive comments! We address your concerns in detail below and have updated our paper according to your suggestions.*
>
> > Q1: the paper does not present convincing arguments/experiments on why the agent would need precise spatial memory for high level semantic goals.
>
> We apologize for the confusion. In our problem formulation—embodied lifelong learning in an open and social 3D world—precise spatial memory is essential because the agent must **execute** its high-level goals, not merely propose them. Precise spatial memory is crucial for providing contexts for the agent's planning, reaction, communication and navigation. Without spatial memory, the agent cannot, for example:
>
> - plan a reasonable daily schedule (e.g., sequence activities to avoid cross-city travel),
>
> - execute low-level actions like navigating to a specific store or picking up an item,
>
> - recall spatial facts when conversing with another agent (as shown in Fig. 4),
>
> - or evaluate whether proposed decisions are feasible given spatial constraints.
>
> Thus, spatial memory is a core component enabling high-level semantic goals to be **grounded and actionable**.
>
> > Q2: The details on how the agent acts to generate low level actions is missing (Are they even generated, e.g. when the agent generates a goal “go to office”, how is this implemented?)
>
> Thank you for raising this. We revised section 4.3.1 and Appendix B to clarify the execution pipeline. In short, when *no reactions are required*, the agent **decomposes each planned activity into finer-grained subgoals** according to activity type and retrieved memory. These subgoals are grounded into **low-level actions** using a combination of navigation algorithms (A*) over voxelized spatial memory and foundation-model-based generation when more complex behaviors are needed. We have expanded these implementation details to make the process clear.
>
> > Q3: The experimental section is also quite week, there aren’t any ablations
>
> Thank you for bringing up the issue. We ablate on the perception challenge with Oracle Perception setting in the paper. Per your suggestion, we've added **new ablation studies** in Table.3 and section 5.2 in the updated paper. We examine four variants on the Detroit community:
> - add a criterion of *importance* during retrieval
> - remove spatial information and the criterion of *spatial proximity*
> - remove multimodal data in the episodic memory image by only calculating text embedding similarity for *content relevance* during retrieval
> - remove *temporal recency* during retrieval
>
> | Setting                | Influence Battle | Leadership Quest |
> |------------------------|------------------|------------------|
> | Full model (ours)      | 46.7             | 37.5           |
> | + *importance*         | 46.7             | 33.3           |
> | - *spatial proximity*  | 40.0             | 25.0           |
> | - *multimodal data*    | 33.3             | 33.3           |
> | - *temporal recency*   | 33.3             | 28.2           |
>
> These ablations show that all parts contribute meaningfully to performance.
>
> >Q4: Some of the experiment settings e.g. (need of robust perception for embodied social agents) are not motivated at all, instead results are presented directly.
>
> We appreciate the chance to clarify. The Oracle Perception setting isolates the **perception challenge** to show how much of the difficulty arises from imperfect 3D perception vs. long-term memory or planning. This is a standard practice in embodied AI benchmarks ([1–3]). We have improved the motivation in the revised text.

---

> ### Author Response · Authors · 2025-11-24
> **Response to Reviewer MVQs [2/2]**
>
> >Q5: It is not clear how the spatial memory is being used by the agent. The paper mentions that the agent uses/needs spatial memory to act, but what is the action here? Are these navigation actions, manipulation actions or other high level semantic actions. If the latter then why do we need spatial memory.
>
> We're sorry for the confusion. The spatial memory, as part of the long-term memory system, will be used to provide context for agent's planning, reaction, communication and navigation.
>
> - Planning: Spatial layout informs which activities can be done sequentially, and at what cost (e.g., having lunch at some place near the office for lunch break).
>
> - Navigation: Low-level waypoint generation relies on the voxelized memory of the environment.
>
> - Interaction: Picking up an object or meeting someone requires knowing their precise or approximate location.
>
> - Communication: Agents reference past locations during conversations (e.g., Fig. 4).
>
> Without spatial memory, the agent cannot reliably ground its high-level decisions or carry out low-level actions.
>
> > Q6: Unclear why precise spatial memory (e.g. which object is at what location) is needed in the future. If an agent goes from place A to place B, then the spatial locations of object in place A (spatial memory of A) is not really needed at place B.
>
> This is a great question. Spatial memory of past locations remains important for:
> - **Conversation.** Agents frequently discuss events, people, or places located elsewhere (Fig. 4b). Accurate spatial memory supports answering questions like “Where did we find the beverages last time?”
> - **Future revists**. Agents commonly revisit locations. Spatial memory prevents full re-exploration from scratch (e.g., locating specific objects faster in the store).
> - **Efficient planning.** Knowledge of where resources were previously seen helps the agent choose where to go next.
>
> We agree that extremely low-level memory may be pruned when storage is limited, and our hierarchical memory design supports selective forgetting intuitively. We only retain all levels of spatial information in our experiments since the storage allows and they might be useful in the cases discussed above.
>
> > Q7: The daily schedule of the agent shown in Figure 3 (c) consists of mostly high level goals (e.g. commute to office). How are these high level goals grounded in the environment? It seems relying on memory alone would be insufficient for many problems (e.g. navigation) which would require planning based on current context instead of past context?
>
> Thanks for this valuable suggestion! To clarify, our long-term memory is updated at every step, and thus includes current context. High level goals are grounded via the planning and reaction framework, where high-level goal is decomposed into subgoals and then turned to low-level actions. Navigation is performed using A* algorithm over the voxel grids stored in the memory, which is described in Appendix.B.1. Thus, planning uses the **latest** state, not outdated or static memory.
>
>
> > Q8: The paper has a very very ambitious goal, to achieve memory for really really long horizon actions (e.g. at the level of days) when the environment clock is ticking at 1 FPS. Does this mean that the number of actions the agent takes in the environment is 1.5 days * 86400? Or does the paper assume some hierarchical actions.
>
> Thank you for raising this! Yes, the environment runs at 1 Hz, so the agent receives perception and control signals at this frequency. However, the decision frequency might be lower: for example, when there is no new objects of interest being perceived, the agent might continue sitting on a sofa for 30 seconds without giving new controls.
>
> [1] Building Cooperative Embodied Agents Modularly with Large Language Models. ICLR 2024
>
> [2] HAZARD Challenge: Embodied Decision Making in Dynamically Changing Environments. ICLR 2024
>
> [3] PARTNR: A Benchmark for Planning and Reasoning in Embodied Multi-agent Tasks. ICLR 2025
>
> *We sincerely appreciate your constructive suggestions to strengthen our work, and hope the new clarifications, expanded explanations, and added experiments address your concerns and turned your assesment positive. Please feel free to let us know if you have further questions.*
>
>
> Best,
>
> Authors

---

### Official Review · Reviewer_pELp · 2025-11-07

**Soundness:** 3
**Presentation:** 4
**Contribution:** 2
**Rating:** 4
**Confidence:** 4

**Summary:**

The authors propose an approach for lifelong multimodal embodied reasoning over long contexts (1.5 days) in the open-world by ingesting visual observations and social interactions. The proposed method implements two different forms of non-parametric memory: a graph-based representation for semantic memory and spatiotemporal episodic memory, in addition to retrieval and reactionary modules (environment interaction and social conversations). The method achieves outperforms adapted baselines on two hand designed social interaction settings testing perception, memory retrieval, reasoning and conversational capabilities.

**Strengths:**

- The paper considers a novel setting of open-world reasoning and social coordination over long contexts (1.5 days).
- The tasks of agent persuasion and team-based task completion considered in the paper are challenging and interesting.
- The accompanying task illustrations greatly improve the understanding of tasks and method performance.

**Weaknesses:**

1. The method appears to be a multimodal extension of prior works: two distinct forms of memory from CoELA and distinct retrieval and reactionary pipelines from Generative Agents, limiting the technical contributions of the proposed method.
2. While the method adds multiple changes over prior works and compares to them, it doesn't carefully study the contribution of each of these changes to the overall improvements. Some of these changes are: name-based graph representation for semantic memory, multimodal data in episodic memory, spatial information in episodic memory, replacing importance with location in relevance computation. These studies would show the importance of these additions to the method.
3. The authors do not report results on standard benchmarks, e.g., TDW-MAT or C-WAH from CoeLA (possibly with oracle perception).
4. Some of the choices seem to be non-intuitive or possibly hand-designed for solving the target tasks.

    a. Same weight to recency as semantics or spatial closeness when calculating may not be generally appropriate: a query could specifically be interested in memories from distant past.

    b. Using closeness of visuals for identifying dynamic objects may not be sufficient, e.g., for dealing with multiple instances of objects. It may be necessary to incorporate further reasoning based on object semantics and temporal distance between the two spottings may be necessary.

     c. Semantic matching to memory matches only images in query to images and texts in query to texts in memory. It may be necessary to incorporate cross-modal matching (e.g., if a memory entry only includes descriptions of objects).

5. Storing entities seen over multiple days to the graph semantic memory could drastically increase the memory size with further increase in number of days. The agent may need to selectively forget memories to avoid large memory blowups.

**Questions:**

1. Are the tasks of Leadership Quest and Influence Battle inspired from prior works? Can authors share more details on the choice of these tasks?
2. Can authors clarify the novel contributions to the method over prior work beyond storing multimodal data and using active perception in place of oracle perception components?
3. Results: Why is Leadership Quest performance for Detroit (or Influence Battle for London) worse with access to oracle perception?
4. Results: Is there evidence to show the inefficiency with oracle perception is actually because of more interactions between agents? Conversely, does the non-oracle approach frequently miss identifying known agents?
5. What are the remaining failures on these settings? Are there limits to the maximal achievable performance on the tasks (e.g., it might be infeasible to persuade all agents for the party in the given time limit if they are initialized very far)?
6. How are the prompts designed for each of the two tasks and how do they differ between the tasks? How much prompt engineering is required to specify goals for solving these tasks?

---

> ### Author Response · Authors · 2025-11-24
> **Response to Reviewer pELp [1/3]**
>
> *We appreciate the insightful and constructive suggestions from you! We address your concerns in detail below and have updated our paper according to your suggestions.*
>
> > Q1: The technical contributions of the proposed method over CoELA and Generative Agents
>
> Thank you for raising this important question. Our work focuses on embodied lifelong learning in an open 3D social world, and introduces several technical contributions beyond both:
>
> - First long-term memory system for embodied agents learning from BOTH visual observations and social interactions.
>     - Generative Agents maintain text-only temporal memory in a 2D symbolic grid world and assume oracle perception.
>     - CoELA contains only short-term, task-specific memory in a confined indoor setting.
>     - In contrast, Ella introduces a dual-form non-parametric long-term memory consisting of Name-centric semantic memory with hierarchical scene graph for spatial knowledge and Spatiotemporal episodic memory that stores images, text, timestamps, and 3D locations.
> - Novel multimodal spatiotemporal retrieval mechanism
>     - Prior work does not retrieve memories using spatial, temporal, and multimodal similarity jointly. Ella’s retrieval ranks all events via spatial proximity, multi-modal content relevance, and temporal recency. This is necessary in an open 600m×600m 3D world, unlike Generative Agent’s purely text-based retrieval or CoELA’s short-horizon access to recent messages only.
> - Structured planning and social interaction grounded in real 3D scene
>     - Generative Agents does not consider actual navigation costs in 3D world, and assumes two agents knowing each other could only engage in a conversation when situated in the same grid for 3 rounds.
>     - CoELA assumes two already-known agents could converse with each other anytime anywhere with task-specific short-term memory.
>     - Ella uses a planning–reaction framework that accounts for 3D navigation cost, visual identification of other agents, and distance-based communication feasibility.
>
> > Q2: While the method adds multiple changes over prior works and compares to them, it doesn't carefully study the contribution of each of these changes to the overall improvements.
>
> Thanks for the detailed and actionable suggestions to improve our work! We have added **new ablation studies** in Table.3 and section 5.2 in the updated paper. We examine four variants on the Detroit community:
> - add a criterion of *importance* during retrieval
> - remove spatial information and the criterion of *spatial proximity*
> - remove multimodal data in the episodic memory image by only calculating text embedding similarity for *content relevence* during retrieval
> - remove *temporal recency* during retrieval
>
> | Setting                | Influence Battle | Leadership Quest |
> |------------------------|------------------|------------------|
> | Full model (ours)      | 46.7             | 37.5             |
> | + *importance*         | 46.7             | 33.3             |
> | - *spatial proximity*  | 40.0             | 25.0             |
> | - *multimodal data*    | 33.3             | 33.3             |
> | - *temporal recency*   | 33.3             | 28.2
>
> These ablations show that all parts contribute meaningfully to performance.
>
> > Q3: Results on standard benchmarks
>
> Thanks for bringing up this issue. To the best of our knowledge, our work is the first to study embodied lifelong learning problem in an open 3D world with social interactions. Prior benchmarks including TDW-MAT and C-WAH only have 2-4 agents without personality, constrained indoor scenes and short temporal horizon, and are not well-formed for testing long-term memories. The closest evaluation platform to us is the one used to test Generative Agemts, though it's only text-based and has unrealistic setting of engaging in conversation automatically and last for at most 3 rounds.
> > Q4 (a): Same weight to recency as semantics or spatial closeness when calculating may not be generally appropriate: a query could specifically be interested in memories from distant past.
>
> Thanks for the valuable question. We agree that there might not be a universally good set of weights for these criteria, and it's the challenging part of building retrieval based memory systems. One promising direction we're actively exploring is to let foudation models adaptively generate these weights given a query, and might even further improve it by post-training with RL techniques.

---

> > ### Author Response · Authors · 2025-11-24
> > **Response to Reviewer pELp [2/3]**
> >
> > > Q4 (b): Using closeness of visuals for identifying dynamic objects may not be sufficient, e.g., for dealing with multiple instances of objects. It may be necessary to incorporate further reasoning based on object semantics and temporal distance between the two spottings.
> >
> > Thank you for the insightful suggestion. This is an excellent point and a real challenge in open-world multi-agent environments. We agree that relying solely on visual similarity can be insufficient in scenarios with multiple visually identical dynamic objects. In our current setting, each agent has a unique visual appearance, which makes appearance-based matching a reliable mechanism for associating detections across frames. However, we acknowledge that in more complex environments—where different agents or objects may share similar visual signatures—additional cues such as semantic attributes, motion consistency, or lightweight temporal tracking would be necessary to maintain robust identity over time. Incorporating such reasoning modules on top of sparse detections is a valuable direction for future extensions of our system.
> >
> > > Q4 \(c):It may be necessary to incorporate cross-modal matching (e.g., if a memory entry only includes descriptions of objects.
> >
> > Thanks for the valuable suggestion. Our method already supports multimodal episodic memory (RGB + text), but we agree that richer cross-modal matching mechanisms would further improve robustness when memory entries contain only partial modalities. We highlight this as an exciting future improvement.
> >
> > > Q5: Storing entities seen over multiple days to the graph semantic memory could drastically increase the memory size with further increase in number of days. The agent may need to selectively forget memories to avoid large memory blowups.
> >
> > Thanks for the valuable suggestion. We agree forgetting is an important direction to work on. As we showed in Figure 5(b), our proposed memory system is more structural, and more efficient than the ones in previous work like generative agents. Our heirarchical design naturally supports pruning of low-level spatial details when memory grows too large. We have added a discussion of future selective-forgetting strategies.
> >
> > > Q6: Are the tasks of Leadership Quest and Influence Battle inspired from prior works? Can authors share more details on the choice of these tasks?
> >
> > Thank you for the valuable question. We select these tasks because they probe meaningful social-cognitive capabilities that require long-term memory, social reasoning, and spatial grounding—rather than narrowly defined short-term behaviors. *Influence Battle* is inspired by the line of research on multi-agent influence and persuasion in mixed-motive environments[1][2], *Leadership Quest* builds on longstanding work in multi-agent coordination and leadership assignment [3][4].
> >
> > > Q7: Can authors clarify the novel contributions to the method over prior work beyond storing multimodal data and using active perception in place of oracle perception components?
> >
> > Thank you for raising this important question. Our work focuses on embodied lifelong learning in an open 3D social world, and introduces several technical contributions over prior work:
> >
> > - First long-term memory system for embodied agents learning from BOTH visual observations and social interactions.
> >     - Generative Agents maintain text-only temporal memory in a 2D symbolic grid world and assume oracle perception.
> >     - CoELA contains only short-term, task-specific memory in a confined indoor setting.
> >     - In contrast, Ella introduces a dual-form non-parametric long-term memory consisting of Name-centric semantic memory with hierarchical scene graph for spatial knowledge and Spatiotemporal episodic memory that stores images, text, timestamps, and 3D locations.
> > - Novel multimodal spatiotemporal retrieval mechanism
> >     - Prior work does not retrieve memories using spatial, temporal, and multimodal similarity jointly. Ella’s retrieval ranks all events via spatial proximity, multi-modal content relevance, and temporal recency. This is necessary in an open 600m×600m 3D world, unlike GA’s purely text-based retrieval or CoELA’s short-horizon access to recent messages only.
> > - Structured planning and social interaction grounded in real 3D geometry
> >     - Generative Agents does not consider actual navigation costs in 3D world, and assumes two agents knowing each other could only engage in a conversation when situated in the same grid for 3 rounds.
> >     - CoELA assumes two already-known agents could converse with each other anytime anywhere with task-specific short-term memory.
> >     - Ella uses a planning–reaction framework that accounts for 3D navigation cost, visual identification of other agents, and distance-based communication feasibility.

---

> ### Author Response · Authors · 2025-11-24
> **Response to Reviewer pELp [3/3]**
>
> > Q8: Why is Leadership Quest performance for Detroit (or Influence Battle for London) worse with access to oracle perception? Is there evidence to show the inefficiency with oracle perception is actually because of more interactions between agents? Conversely, does the non-oracle approach frequently miss identifying known agents?
>
> Thanks for pointing this out. With oracle perception, Ella identifies other agents and objects more easily and confidently, leading to more intense competitions between different groups and quite different social dynamics. Nevertheless, Ella still performs better than all other baselines after removing the perception challenge.
>
> > Q9: What are the remaining failures on these settings? Are there limits to the maximal achievable performance on the tasks (e.g., it might be infeasible to persuade all agents for the party in the given time limit if they are initialized very far)?
>
> Thanks for the valuable question! Theoretical ceiling performance on these two tasks should be 100.0. Considering all communities span 600m * 600m, agents can reach any place on foot within an hour (3600 low-level actions) in all the communities, and there are more than enough target items if considering all stores. Failures primarily arise from insufficient cooperation, poor communication of discovered information, and the uncertainty of the environments.
>
>
> > Q10: How are the prompts designed for each of the two tasks and how do they differ between the tasks? How much prompt engineering is required to specify goals for solving these tasks?
>
> Thank you for the valuable question! We fully agree that minimal prompt engineering is desirable to specify goals for different tasks. As described in Lines 177–178, during controlled evaluation, the only intervention is **changing the community-level goals**; no task-specific prompt tuning is used. We emphasize this design principle more clearly and include the prompt for thw two tasks in Appendix.A.2 and Figures 9-10.
>
>
> *[1] Social Influence as Intrinsic Motivation for Multi-Agent Deep Reinforcement Learning. ICML 2019*
>
> *[2] Persuasion, influence, and value: Perspectives from communication and social neuroscience. Annual Review of Psychology 2018*
>
> *[3] Empowering leadership in management teams: effects on knowledge sharing, efficacy, and performance. Academy of Management Journal 2006*
>
> *[4] Embodied LLM Agents Learn to Cooperate in Organized Teams. NeurIPS 2024*
>
>
> *We sincerely appreciate your insightful and constructive suggestions, and hope our new experiments and clarifications help address your concerns and turned your assesment positive. Please feel free to let us know if you have further questions.*
>
> Best,
>
> Authors

---

### Author Response · Authors · 2025-12-01
**General Response to All Reviewers and ACs [1/2]**

We thank all the reviewers and ACs for their time and effort in reviewing our paper and giving insightful comments. In addition to our detailed responses to specific reviewers, we would like to highlight our contributions and clarify some concerns.

**1. Our Contributions**

We are glad to find that the reviewers have acknowledged our following contributions:

- **Tackles a novel and important problem of embodied lifelong learning in an open and social 3D world with a structured non-parametric long-term memory**
    - a novel setting of open-world reasoning and social coordination over long contexts (1.5 days). [reviewer pELp]
    - a very ambitious problem of learning in situated agents with non-parametric memory. Also a very important and challenging problem. [reviewer MVQs]
    - This paper introduce a novel and grounded memory architecture [reviewer j9Ji]
    - The paper tackles an ambitious and important problem - enabling embodied agents to learn and operate over extended temporal scales (days) in social environments. I found it a fun read, and there were plenty of good cogsci + psychology references. [reviewer k6G2]


- **Well-designed experiments and in-depth analysis to provide insights**
    - The tasks of agent persuasion and team-based task completion considered in the paper are challenging and interesting. [reviewer pELp]
    - a clear and significant performance advantage. The analysis effectively pinpoints the baselines' failures. This paper also includes thoughtful analysis which is a good base for future work [reviewer j9Ji]
    - The evaluation setup is creative and goes beyond typical navigation or manipulation tasks.[reviewer k6G2]
    - The paper includes ablation studies with oracle perception, showing the impact of perception errors on the overall system performance. [reviewer k6G2]
    - The inclusion of experiments with open-source foundation models (DeepSeek-R1, Qwen2.5) shows the system depends heavily on the LLM used [reviewer k6G2]

**2. Clarifications**

- **The focus of the paper**
    - Our setting is fundamentally different from (a) a multi-agent setting with oracle perception and (b) a single-agent embodied perception setting.
    - Our problem formulation targets embodied lifelong learning in an **open, social 3D world**, where agents accumulate knowledge from **both visual perception and social interactions**. This dual modality of experience is essential, humans continuously learn from visual perceptions and social interactions, and existing long-term memory systems do not support this unified learning process.
    - To the best of our knowledge, we're the first to build such memory system that can learn from **both visual observations and social interactions**.

- **The technical contributions of the proposed method over Generative Agents and CoELA**
    - Ella is the first long-term memory system for embodied agents learning from **BOTH visual observations and social interactions**.
        - Generative Agents maintain **text-only temporal** memory in a **2D symbolic grid world** and assume oracle perception.
        - CoELA contains only **short-term, task-specific** memory in a confined **indoor setting**.
        - In contrast, Ella introduces a dual-form non-parametric long-term memory consisting of Name-centric semantic memory with hierarchical scene graph for spatial knowledge and Spatiotemporal episodic memory that stores images, text, timestamps, and 3D locations.
    - Novel multimodal spatiotemporal retrieval mechanism
        - Generative Agents's retrieval mechanism neither considers spatial information nor multi-modal information.
        - CoELA has short-horizon access to recent messages only.
        - Ella’s retrieval ranks all events via **spatial proximity**, **multi-modal content relevance**, and **temporal recency**. This is necessary in an open and social 3D world, the one humans and future embodied agents inhabit.
    - Structured planning and social interaction grounded in real 3D geometry
        - Generative Agents does not consider **actual navigation costs in 3D world**, and assumes two agents knowing each other could only engage in a conversation when situated in the same grid for pre-set 3 rounds.
        - CoELA assumes two already-known agents could converse with each other **anytime anywhere** with task-specific short-term memory.
        - Ella uses a planning–reaction framework that accounts for 3D navigation cost, **visual identification of other agents**, and **distance-based communication feasibility**.

---

> ### Author Response · Authors · 2025-12-01
> **General Response to All Reviewers and ACs [2/2]**
>
> **Revision Summary**
>
> - We have added **new ablation studies** following reviewer **pELp**, **MVQs** and **k6G2** 's suggestion, and the results show that all parts contribute meaningfully to performance. We include the results and discussions in Table 3 and Section 5.2 in the revised paper. We examine four variants on the Detroit community:
>     - add a criterion of *importance* during retrieval
>     - remove spatial information and the criterion of *spatial proximity*
>     - remove multimodal data in the episodic memory image by only calculating text embedding similarity for *content relevance* during retrieval
>     - remove *temporal recency* during retrieval
>
>     | Setting                | Influence Battle | Leadership Quest |
>     |------------------------|------------------|------------------|
>     | Full model (ours)      | 46.7             | 37.5             |
>     | + *importance*         | 46.7             | 33.3             |
>     | - *spatial proximity*  | 40.0             | 25.0             |
>     | - *multimodal data*    | 33.3             | 33.3             |
>     | - *temporal recency*   | 33.3             | 28.2
>
> - We add additional details in section 4.3.1 and Appendix B to clarify the execution pipeline as Reviewer **MVQs** suggested.
>     - In short, when no reactions are required, the agent decomposes each planned activity into finer-grained subgoals according to activity type and retrieved memory. These subgoals are grounded into low-level actions using a combination of navigation algorithms (A*) over voxelized spatial memory and foundation-model-based generation when more complex behaviors are needed.
> - We explain the motivation more clearly in section 5.2 as Reviewer **MVQs** suggested.
> - We add more details on the task selection in Appendix A.2 to explain the motivation and connect with prior literature more clearly as Reviewer **pELp** and **k6G2** suggested.
> - We include the prompt for the two tasks in Appendix.A.2 and Figures 9-10 to emphasize the design principle of minimal prompt engineering to specify goals for different tasks more clearly as Reviewer **pELp** suggested.
> - We add an analysis of perception-error propagation in Appendix A.4 as Reviewer **j9Ji** suggested.
> - We add a paragraph to discuss the scalability of our memory system design and future selective-forgetting strategies in Appendix A.4 as Reviewer **pELp** and **j9Ji** suggested.
> - We add a paragraph to clarify the name overlap with "ELLA: Efficient Lifelong Learning Algorithm" in Appendix.A.4 as Reviewer **k6G2** suggested.
>     - We chose the name for its semantic meaning (“her” in Spanish), reflecting our long-term vision for embodied agents learning socially and visually.
>
> We hope our detailed responses below convincingly address all reviewers’ questions.

---

### Meta-Review · Area_Chair_jYDV · 2025-12-22

**Summary:**

The paper mostly received reject ratings from the reviewers (6,4,2,2). The reviewers mentioned various concerns such as:
- Lack of ablation experiments
- No results on the standard benchmarks
- Memory blowup with longer horizons
- Feature similarity limiting the reasoning to simple tasks
- A limited multi-agent setup on two hand-defined tasks in a single hand-defined setting
- Involving a handcrafted retrieval rule, similar to conceptgraphs, that builds on feature similarity and spatiotemporal recency.

The rebuttal addresses some of these concerns. For example, a set of ablation experiments are provided to show the effectiveness of each design choice and it is clarified that this is a new task with no existing benchmarks. The problem itself is interesting and potentially impactful, but the current formulation and solution approach do not yet make a strong case for the paper’s overall contribution. The method largely combines multiple modules without an overarching optimization objective, and the evaluation is limited to a single environment. As a result, it remains unclear what type of generalization is being achieved and how the approach would extend to new environments or tasks. Also, some of the arguments in the rebuttal are not convincing. For instance, the existing scene graph models can be easily extended to this task. Due to these issues, the AC is in agreement with the majority of the reviewers and recommends rejection.

**Reviewer Concerns:**

Please refer to the box above.

**Reviewer Scores:**

Reviewer pELp: The technical contribution aspect of the paper is weak so this reviewer would not change the score.

Reviewer MVQs: The reviewer expected a major rewrite since they found the paper confusing due to lack of details. The rebuttal provided some clarification, but there is still a lot of room for improvement. No change of score would be expected.

Reviewer j9Ji: No change would be expected as well since the authors leave complex queries for future work.

Reviewer k6G2: This reviewer would probably increase the score to 4 since some issues were addressed, but they are still not convinced by the handcrafted rules for memory.

---

### Decision · Program_Chairs · 2026-01-26

Reject